# Causal Fine-Tuning under Latent Confounded Shift

Jialin Yu[1]   Yuxiang Zhou[2,3]   Haoxuan Li[4]   Junchi Yu[1]   Mengyue Yang[5]   Yulan He[3]   Nevin Zhang[6]
Philip Torr[1]   Ricardo Silva[7]

## Abstract

Adapting to latent confounded shift remains a core challenge in modern AI. This setting is driven by hidden variables that induce spurious correlations between inputs and outputs during training, leading models to rely on non-causal shortcuts. For example, a model may learn to treat metadata (e.g., data source like "Amazon") as a proxy for positive sentiment, causing failure when the source becomes predominantly negative during deployment. To address this *latent confounded shift*, we introduce Causal Fine-Tuning (CFT). Using a structural causal model as an inductive bias, we derive sufficient identification conditions that motivate a fine-tuning objective for decomposing representations into high-level stable and low-level shift-sensitive components. Instantiating this framework in BERT, we show that learning such causal/spurious representations and adjusting them accordingly yield a more robust predictor. Experiments on spurious correlation injection attacks in text demonstrate that our method outperforms black-box domain generalization baselines, highlighting the benefits of explicitly modeling causal structure[1].

## 1. Introduction

Distribution shift is a fundamental challenge for many modern machine learning systems (Huang et al., 2006; D'Amour et al., 2022). In practice, such shift is often implicit and confounded (Veitch et al., 2021; Alabdulmohsin et al., 2023),

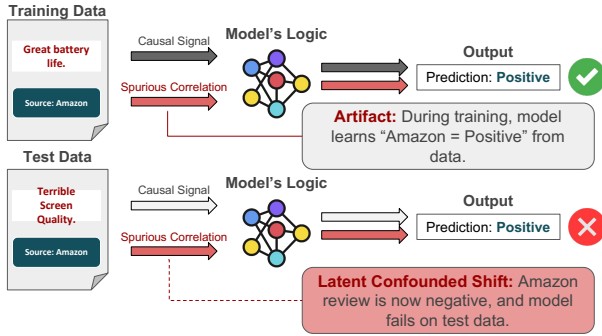

*Figure 1.* An illustration of a latent confounded shift: the spurious correlation between data source and sentiment label flips at test time.

where latent variables induce spurious correlations between inputs and outputs. We refer to this broader class of distribution shift, arising either directly at the latent confounder level or through interactions involving latent confounders, as *latent confounded shift*[2]. Examples of such shift include deploying models across different hospital populations (Caruana et al., 2015), handling corrupted and adversarial image (Hendrycks & Dietterich, 2019; Ilyas et al., 2019), stress testing models (D'Amour et al., 2022), and syntactic shortcuts in post-trained language models (Shaib et al., 2025).

To address such concerns, previous work has explored feature augmentation and regularization (Hendrycks et al., 2019; Xie et al., 2020; Zhang et al., 2020b; Tu et al., 2020), as well as learning invariant features across domains (Arjovsky et al., 2019; Ahuja et al., 2020; Heinze-Deml & Meinshausen, 2021). Although effective in certain settings, these methods typically assume stable invariant features are the only useful signals across environments. However, they do not explicitly account for scenarios where observed features can entangle causal signal with spurious correlations that vary across environments. In Figure 1, we illustrate with an example where the model learns an implicit spurious correlation between Amazon and positive label from training data, but this correlation flips in test data (Amazon

---

[1]University of Oxford, United Kingdom [2]Queen Mary University of London, United Kingdom [3]King's College London, United Kingdom [4]Peking University, China [5]University of Bristol, United Kingdom [6]Hong Kong University of Science and Technology, China [7]University College London, United Kingdom. Correspondence to: Jialin Yu <yu.jialin@outlook.com>.

*Proceedings of the 43$^{rd}$ International Conference on Machine Learning*, Seoul, South Korea. PMLR 306, 2026. Copyright 2026 by the author(s).

[1]Our code and data are released at https://github.com/jialin-yu/CausalFineTuning.

[2]This generalizes the notion of "latent confounder shift" to include cases where the latent confounder itself may not shift, but its interactions with other variables induce distributional changes.

reviews become mostly negative). Such spurious correlations are not hypothetical: platform identifiers, annotation artifacts, and dataset collection heuristics often create precisely this kind of spurious linkage in practice (Gururangan et al., 2018; Sagawa et al., 2019).

Foundation models (Bommasani et al., 2021), such as BERT (Kenton & Toutanova, 2019), GPT (Brown et al., 2020), and CLIP (Radford et al., 2021), have become the standard for downstream learning. However, when fine-tuned on data with latent confounding, they often exploit spurious correlations that do not generalize across environments (Lv et al., 2022; Qiao & Low, 2024). In this paper, we address this issue by proposing a framework for fine-tuning under latent-confounded shifts, using causal identification as a design principle for constructing robust inductive biases. We define *causal fine-tuning* as the process of learning a predictor that targets the causal mechanism (rather than spurious correlations). Specifically, this is achieved by encouraging a representation decomposition that is consistent with an identification-motivated latent structure, separating fine-tuned representations into stable (invariant) and spurious (environment-specific) components. Our key insight is to exploit scenarios where it can be assumed that: i) *low-level* features indicate how the input distribution shifts across data regimes; ii) they causally influence both *high-level* features and the label, with the former possibly changing in distribution at test time. Unlike standard covariate-shift scenarios, the label can change indirectly through shifts in latent confounders. By applying a causal adjustment strategy to these representations, leveraging the two views of the unstable features (unsupervised and fine-tuned), we produce robust and adaptive predictions without requiring multiple environments or domain annotations.

Our contributions are as follows. (1) **Modeling:** we introduce an identification scaffold and derive sufficient conditions that motivate a robust fine-tuning objective (Section 4); (2) **Algorithm:** we translate this identification-guided structure into an inductive bias for learning, yielding a practical method that decomposes fine-tuned representations into invariant and spurious components within standard fine-tuning pipelines (Section 5); (3) **Empirical results:** we demonstrate improved generalization under latent confounded-shift with spurious correlation injection attacks to mimic deployment artifacts (Section 6); and (4) **Ablation:** we analyze the role of key components in the algorithm and their impact on robustness and predictive performance (Section 6).

## 2. Related Work

Distribution shift is fundamentally an ill-posed problem without assumptions: the mapping between input and labels may change arbitrarily across domains (Huang et al., 2006). A central question is: which parts of the data-generating process remain invariant across environments? Causal inference, in particular transportability theory (Pearl & Bareinboim, 2011; Jalaldoust & Bareinboim, 2024), provides a principled framework for answering this by characterizing how and when causal knowledge can be moved between environments. The most common assumptions in the literature are covariate shift (Magliacane et al., 2018; Shimodaira, 2000) and label shift (Buck & Gart, 1967). Building on these assumptions, some lines of work explore methods that learn causally robust invariant features $\Phi(x)$ based on the covariates across multiple environments (Arjovsky et al., 2019; Ahuja et al., 2020; Von Kügelgen et al., 2021; Yue et al., 2021; Mitrovic et al., 2021; Shi et al., 2021; Kong et al., 2023). Another line of work explores counterfactual reasoning and data augmentation techniques (Kaushik et al., 2019; Ben-David et al., 2022; Le et al., 2023; Feder et al., 2023; Zhou & Zhu, 2025; Han et al., 2025).

More recently, there has been a growing interest in causal representation learning, which aims to learn disentangled latent representations that capture the underlying causal structure. Existing work focuses on learning stable causal latent variables (Sun et al., 2021; Lu et al., 2022; Lv et al., 2022), invariant predictors (Veitch et al., 2021; Jiang & Veitch, 2022), or compositional models (Bravo-Hermsdorff et al., 2023; Yu et al., 2024). These approaches often require multiple environments or proxy variables, which can be impractical to acquire or augment, particularly for unstructured data (Chalupka et al., 2017). We build on this body of work by treating a pre-trained foundation model as an implicit environment for data augmentation, on top of the fine-tuning data, allowing causal representation learning with single domain fine-tuning data only. Once the latent variables are learned, causal adjustment formulas are adopted to generate predictors that are robust to domain shift, with two common receipts being back-door adjustments (Yue et al., 2020; Zhang et al., 2020a), and front-door adjustments (Li et al., 2021; Mao et al., 2022; Nguyen et al., 2023; Li et al., 2025). Front-door adjustments have become a promising choice due to hidden confounding variables between input signals and outputs. Motivated by the recent success of these methods, we propose a fine-tuning procedure that implements a front-door–style causal adjustment, achieving improved OOD robustness under confounded shifts.

## 3. Preliminaries

**Motivation.** Our proposal encodes invariance assumptions into structural causal models (SCMs) with explicit regime indicator variables (Pearl, 2009; Dawid, 2021). In particular, we consider the scenario where input $X$ is allowed to cause label $Y$, but not vice-versa, with the possible presence of hidden confounders $U$. Graphically, this setup is depicted in Figure 2(a). To accommodate distribution shifts,

we further assume that changes from training to test environments involve an *intervention* (or *perturbation*, or *regime*), denoted by the *regime variable* $\sigma$ (Dawid, 2021), which modifies the influence of $U$ on $X$. Data are observed only under the training regime denoted by $\sigma = $ train. We want to be safe to build a predictor for an unknown environment denoted as $\sigma = $ test, where the conditional distribution $p(x \mid u; \sigma = $ test$)$ can potentially arbitrarily differ from $p(x \mid u; \sigma = $ train$)$.

**Intuition and principles.** Let us first explicitly analyze why distribution shifts under our postulated structure lead to machine learning classifiers failing.

**Proposition 3.1.** *Let a causal model, following the structure shown in Fig. 2(a), represent the source (train) and target (test) domains of some probabilistic system resulting from regimes $\sigma = $ train and $\sigma = $ test, with respective implied distributions $p(y \mid x) := p(y \mid x; \sigma = $ train$)$ and $p^\star(y \mid x) := p(y \mid x; \sigma = $ test$)$. Then, in general, $p(y \mid x) \neq p^*(y \mid x)$.* □

This follows directly from the law of total probability over $U$, below assumed to be discrete without loss of generality:

$$p(y \mid x; \sigma) = \sum_u p(y \mid u, x; \sigma)p(u \mid x; \sigma)$$
$$= \sum_u \underbrace{p(y \mid u, x)}_{\text{does not change with } \sigma} \underbrace{p(u \mid x; \sigma)}_{\text{changes with } \sigma}.$$

This implies that a predictor learned under $\sigma = $ train is not transportable (Pearl & Bareinboim, 2011; Jalaldoust & Bareinboim, 2024) in this setting even when $\sigma$ affects neither $U$ nor $Y$ directly. To exploit the postulated causal structure, we therefore focus on the fact that the change from $p(y \mid x; \sigma = $ train$)$ to $p(y \mid x; \sigma = $ test$)$ boils down to the difference between $p(u \mid x; \sigma = $ train$)$ and the unknown $p(u \mid x; \sigma = $ test$)$. One possibility is to address it in a minimax way, akin to distributionally robust optimization (Duchi & Namkoong, 2021), where we minimize our loss with respect to all distributions $p^\star(u \mid x; \sigma = $ test$)$ which are "close" to $p(u \mid x; \sigma = $ train$)$ in some sense. However: i) in general we will not be able to identify $p(u \mid x; \sigma = $ train$)$ nor $p(y \mid x, u)$ from observational data alone; ii) $U$ is latent and typically lacks a measurable proxy, making it difficult to specify a scientifically meaningful notion of closeness between $p^\star(u \mid x; \sigma = $ test$)$ and $p(u \mid x; \sigma = $ train$)$; iii) it may inherit well-known limitations of the minimax approach, being potentially overly conservative when $p(u \mid x; \sigma = $ test$)$ and $p(u \mid x; \sigma = $ train$)$ are close.

**Maximum-entropy as a default choice.** We propose following a *default regime* built from a maximum-entropy distribution $p(u \mid x; \sigma = $ default$)$ that respects the marginal

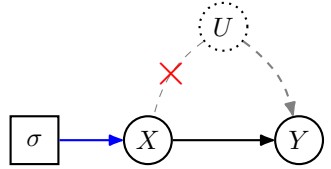

(a) Original structural causal model.

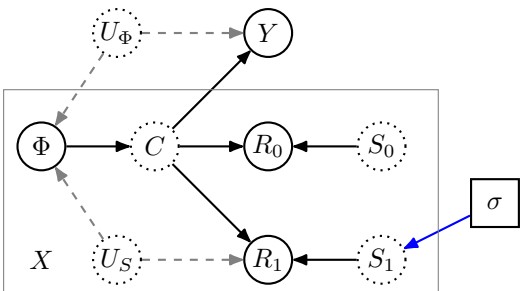

(b) Refinement of the original structural causal model.

*Figure 2.* **(a)** Dashed vertices represent hidden variables and square *regime* vertices represent interventions, perturbations or changes of environment. The graph indicates that the mechanism into $X$ may change according to regimes indexed by a regime variable $\sigma$. For instance, when a $\sigma = $ do$(x)$ operation is performed (Pearl, 2009), the edge between $U$ and $X$ is removed, indicated by a red cross. In general, $\sigma$ indexes arbitrary distributions (Dawid, 2021). **(b)** Identification scaffold used to motivate the CFT objective, where $X$ is broken apart, and abstracted into vectors $R_0$, $R_1$ and $\Phi$ as described in Section 4. More intuition of this causal diagram is discussed in Appendix D.

constraint $p(u; \sigma = $ default$) = p(u; \sigma = $ test$)$. A trivial solution is $p(u \mid x; \sigma = $ default$) = p(u; \sigma = $ test$)$, the latter which is also knowable by the assumption: $p(u; \sigma = $ test$) = p(u; \sigma = $ train$)$. This also means that $p(y \mid x; \sigma = $ default$) = p(y \mid x; \sigma = $ do$(x))$, since our causal structure assumption implies $p(u; \sigma = s^\star) = p(u; \sigma = s^{\star\star})$ and $p(y \mid x, u; \sigma = s^\star) = p(y \mid x, u; \sigma = s^{\star\star})$, for all values $s^\star, s^{\star\star}$ in the scope of $\sigma$.

We thus reduce this to the problem of training our model *as if* it came from the regime $\sigma = $ do$(x)$, but using data collected under $\sigma = $ train. Unfortunately, it is a well-known result that, assuming no more than the Markovian factorization implied by Figure 2(a), the distribution $p(y \mid $ do$(x)) := p(y \mid x; \sigma = $ do$(x))$ is not identifiable: it follows from the completeness of Pearl's do-calculus (Pearl, 2009). To that effect, in the sequel we assume a more fine-grained structure for $X$, as well as making explicit use of fine-tuning data.

## 4. Assumptions for Causal Fine-Tuning

In this section, we introduce the structural assumptions required for identifiability, then discuss their implications for practical applications. Standard supervised learning makes no distinction between "causal features" and "non-

causal features". Moreover, concepts such as $do(x)$ may be ill-posed when the object of intervention is unstructured (Chalupka et al., 2017), such as raw text.

**Scope.** This section focuses on an *abstract* identification analysis: we start by postulating which representations of $X$ are assumed to follow a causal structure. More intuition of this formulation is discussed in Appendix D. The causal graph in Fig. 2(b) provides *sufficient* conditions for identifiability and serves as an identification construct. The SCM below should be read as an identification scaffold rather than a literal data-generating graph for text. We do not claim that natural language literally follows this SCM; rather, it specifies the invariances and (in)dependences that motivate our subsequent modeling choices. The *algorithmic construction* of the measurements used in this model (e.g., how to obtain $(R_0, R_1, \Phi)$ from a foundation model) is deferred to Section 5.

**Assumption 4.1** (**Functional Decomposition**). We assume access to a triplet of measurements $(R_0, R_1, \Phi) = f(X)$ for some function $f$, defined non-constructively, as follows: (i) $(R_0, R_1)$ is a **paired representation** of $X$. The mapping to $R_0$ is learned using self-supervised learning with unlabeled data. The mapping to $R_1$ is learned during supervised fine-tuning with labeled data under the training environment $\sigma =$ train; (ii) **local features** $\Phi$ are low-level features that can be obtained by a learnable mapping (e.g., a small module) from intermediate activations of the fine-tuned model, which summarize environment-dependent cues. $\square$

In the sequel, we will explicitly describe the procedure that constructively defines this mapping $(R_0, R_1, \Phi) = f(X)$. It is to be noted that an intervention $do(x)$ will be defined as $do((R_0, R_1, \Phi) = f(x))$, which we will sometimes denote as $do(R_0), do(R_1), do(\Phi)$.

Under this choice of abstraction, we postulate a causal structure with $(R_0, R_1, \Phi)$ interacting with causal latent variables $C$ and two sets of "spurious" latent variables $S_0$ and $S_1$, spurious in the sense that only $C$ is a causal parent for output $Y$. We define $S_0$ as the latent spurious features of pre-training not affected by distribution shifts, and $S_1$ are the latent spurious features affected by the environment index $\sigma$. The generative model contains these two feature sets as latent variables, along with structural assumptions about how $\sigma$, $R_0$, $R_1$, $\Phi$ and $Y$ are connected. Assumptions are graphically summarized in Fig. 2(b) and detailed as follows.

**Assumption 4.2** (**Causal Latent Structure**). High-level features $\{R_0, R_1\}$ are indirect measurements of mutually independent variables $\{S_0, S_1, C\}$. $S_0$ can only cause $R_0$ and $S_1$ can only cause $R_1$. The regime variable $\sigma$ can only affect $S_1$. Moreover, hidden confounders $U_S$ are common parents of $R_1$ and $\Phi$, and independent hidden confounders $U_\Phi$ are parents of $\Phi$ and $Y$. $\square$

This assumption aligns with prior work in causal learning (Tenenbaum & Freeman, 1996; Gong et al., 2016; Heinze-Deml & Meinshausen, 2021; Mao et al., 2022). Intuitively, this abstracts the true complex causal graph into a coarser granularity, encapsulating stable hidden confounders into $C$ and any other (unstable) non-confounding variables into $S_0, S_1$. It also postulates a principle: *any dependency between $R_0$ and $R_1$ is solely attributed to common cause $C$.*

**Assumption 4.3** (**Causal Structure of Distribution Shifts**). Regime variable $\sigma$ affects the system only via $S_1$. This also implies that the causal ancestors of $Y$ do not interact with $\sigma$. $\square$

This assumption postulates that, for any regime of interest where we deploy our system, the relationship between causal ancestors and the output $Y$ is invariant. However, it is not the case that we will be able to optimize the empirical risk on the training data without consequences, since conditioning on the entire input signal $\{R_0, R_1, \Phi\}$ will *d-connect* $Y$ with $\sigma$ (Pearl, 2009): this happens via *active collider paths* (Spirtes et al., 2000) such as $Y \leftarrow U_\Phi \rightarrow \Phi \leftarrow U_S \rightarrow R_1 \leftarrow S_1 \leftarrow \sigma$ and $Y \leftarrow C \rightarrow R_1 \leftarrow S_1 \leftarrow \sigma$. This makes our predictions dependent on the value of $\sigma$, in the sense of (Dawid, 2021), which means being affected by distribution shifts. In what follows, we will rely on i) the missing edge $\Phi \rightarrow Y$ and ii) the ability of deterministically inferring $C$, as formalized in the following assumption and theorem.

**Assumption 4.4** (**Sufficient Mediator**). The causal effect of $\Phi$ on $Y$ is fully mediated through $C$. That is, $p(y \mid do(\Phi), do(C)) = p(y \mid do(C))$. $\square$

**Justification.** This assumption is sometimes known as a front-door structure (Pearl, 2009) for the effect of $\Phi$ on $Y$. It can be interpreted as having $C$ as ultimately the only variable driving $Y$ directly, and relying on this desiderata as the operational *definition* of $C$, implying no further latent sources confounding $\Phi$ and $C$, or $C$ and $Y$, or any other path between $\Phi$ and $Y$ relying on further (implicit) hidden variables. We allow *observational confounding* between $\Phi$ and $Y$ via $U_\Phi$, while assuming $\Phi$ has no *direct causal effect* on $Y$ once $C$ is intervened on.

**Theorem 4.5** (**Identification for Causal Features** $C$). *Assume the structural assumptions encoded in the causal graph in **Fig. 2(b)**. Let the mapping between $\{S_0, S_1, C\}$ and $\{R_0, R_1, \Phi\}$ obey the invertibility conditions of (Von Kügelgen et al., 2021). According to* Theorem 4.4 *in (Von Kügelgen et al., 2021), we can learn a deterministically function mapping from $R_0$ and $R_1$ to a representation of the stable component $C$, such that $p(C|R_0) \approx p(C|R_1)$.*

**Intuition.** This theorem implies that, under the stated abstraction, if the causal latent variable $C$ remains invariant

across environments (Assumption 4.3), the distribution shift between representations $R_0$ and $R_1$ can be used to learn a proxy for the stable component $C$. For a formal proof of this theorem, please refers to *Theorem 4.4* in (Von Kügelgen et al., 2021). In the sequel, we will learn this function using the idea presented in Equation 4. Note that once we learn this mapping, we just need one of $R_0$ or $R_1$ to estimate $C$. In the algorithm, $C$ should therefore be understood as a learned proxy for the stable component implied by the identification scaffold, not as a recovered ground-truth causal variable.

We will now show that, under this identification scaffold, the interventional target $p(y \mid do(x))$ can be expressed in terms of observable or learned quantities. The proof of this result is short and presented in Appendix I.

**Theorem 4.6** (**Identification for Causal Transfer Learning**). *Given the assumptions in the causal graph in **Fig. 2 (b)** and Theorem 4.5, the distribution of $Y$ under $do(x)$ can be computed as*[3]

$$p(y \mid do(x)) = \sum_{\Phi', x'} p(y \mid \Phi', C) p(\Phi' \mid x') p(x') \quad (1)$$

$$= \mathbb{E}_{x' \sim p(x')} \, \mathbb{E}_{\Phi' \sim p(\Phi'|x')} \left[ p(y \mid \Phi', C) \right], \quad (2)$$

*where $C$ is given as a function of $(R_0, R_1)$ per Theorem 4.5, and $(R_0, R_1)$ are a function of $x$ during training. $C$ is given as a function of $(R_1)$, and $(R_1)$ as a function of $x$ at inference time* $\square$

**Invariance implication and pragmatic application.** The difference between $p(y \mid x; \sigma = \text{test})$ and $p(y \mid x; \sigma = do(x))$ in our setup boils down to averaging $p(y \mid C, U_\Phi)$ over $p(u_\Phi \mid R_0, R_1, \Phi, C; \sigma = \text{test})$ in the former, and $p(U_\Phi)$ in the latter. When can we say that the latter is an improvement over $p(U_\Phi \mid R_0, R_1, \Phi, C; \sigma = \text{train})$? Our claim is that by virtue of the confounder being a cause of local features $\Phi$ only, and not of the whole of $X$, the relevance of information passing through $(S_0, S_1)$ via active collider paths only should be limited anyway (Ding & Miratrix, 2014), *unless the test environment affects it drastically*. In this case, we may be thrown away too far from the original $p(U_\Phi \mid R_0, R_1, \Phi, C; \sigma = \text{train})$ in unpredictable ways, and the safer bet ("maximum-entropy") is to think of $p(y \mid C)$ as being a random measure "$p_{U_\phi}(y \mid C)$" with a conservative prior $p(U_\Phi)$ which comes from the model and is agnostic to the environment. Section 6 details this empirical investigation.

---

[3]$\Phi'$ is deterministically given by $x'$, but the above representation in terms of a probability $p(\Phi' \mid x')$ is useful as a way of understanding how to generate $\Phi'$.

## 5. Algorithm: Causal Fine-Tuning

In this section, we operationalize the theoretical insights outlined in Section 4 into a Causal Fine-Tuning (CFT) framework (illustrated in Fig. 3).

**SCM as an inductive bias.** The causal model in Fig. 2(b) is used as an *identification-guided inductive bias* for designing objectives, not as a claim that the foundation model contains explicit variables $C$ and $\Phi$, nor that we recover the ground-truth SCM. Instead, our algorithm learns *representation-based estimators* of these factors such that their empirical behavior approximately satisfies the assumptions in Section 4. All identification results are therefore interpreted as *motivation and guidance* for the training objective; robustness is ultimately validated empirically. In practice, we only require $C$ to be stable across the two views (frozen vs. tuned) and $\Phi$ to capture shift-sensitive cues; if these properties fail, the method degrades toward standard fine-tuning. To avoid notational clutter, we reuse the symbols $C$ and $\Phi$ to denote these learned estimates whenever it is unambiguous.

**Component 1: Supervised Fine-Tuning** The first step is to learn $R_1$ from training samples through supervised fine-tuning (SFT). Assume for exposition purposes that labels $Y$ are binary. Given a pre-trained model $p(r_0 \mid x)$, we learn $p(r_1 \mid x)$ with training samples $(X, Y)$ by minimizing the loss

$$\mathcal{L}_{\text{SFT}} = \mathbb{E}_{(x,y) \sim \mathcal{D}} \left[ -y \log p(y \mid x) \right]. \quad (3)$$

where $r_1$ is the sentence representation taken from the last layer of the fine-tuned model.

**Component 2: Learning Causal Representation** To learn the invariant causal feature $C$, we aim to identify the distribution $p(c \mid r)$. This process involves aligning representations from different environments while maximizing entropy and preventing collapsed representations (Von Kügelgen et al., 2021). The loss function is constructed based on Theorem 4.5,

$$\mathcal{L}_C := \mathbb{E}_{(r_0, r_1 \mid x) \sim \mathcal{D}} \left[ \| p(c \mid r_0) - p(c \mid r_1) \|_2^2 \right] \\ - H\left( p(c \mid r_0) \right) - H\left( p(c \mid r_1) \right), \quad (4)$$

where $x$ is sampled from $p(x)$ and used to calculate $r_0, r_1$. The first term enforces invariance across environments. The entropy terms maximize the diversity in representations, reducing the risk of collapse.

**Component 3: Learning Local Representation** This component focuses on learning local representation $\Phi$ from the fine-tuned model intermediate activations, in theory any layer of the model can be used to learn $\Phi$. In our running

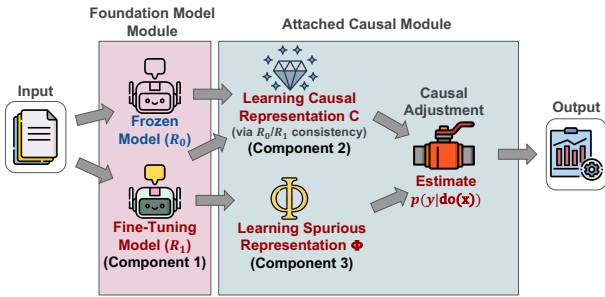

*Figure 3.* Illustration of our CFT methods. During training, we keep a copy of pre-trained foundation model (Frozen Model) for identification purposes, which is removed during inference. Once CFT is done, we get a model of the similar size as the standard fine-tuning but providing functions to decompose input to causal and spurious features. This allows for adaptation to latent confounded shifts at test time.

example, we use the embedding layer, which is similar to the setup in (Mao et al., 2022). This can be further justified that the first layer provides a better low-level signal (LeCun et al., 2015; Goodfellow et al., 2016). See more discussion in Appendix E.

To assess the sensitivity of $\Phi$ extraction, we conduct a layer-sensitivity study across the embedding layer, early transformer layers, intermediate layers, and the last layer (shown in Table 6). The results show that CFT is reasonably stable across layer choices, but the current embedding-layer design is strongest under the more severe shifts, while intermediate and higher-layer choices can remain competitive in weaker-shift regime. This is consistent with our hypothesis that lower-layer features provide a stronger low-level signal for capturing shift-sensitive cues, whereas higher layers increasingly mix in more stable semantic information that is better attributed to $C$.

Given input $X$ as a series of tokens where $X = [t_1, t_2, ..., t_m]$, we can retrieve a vector representation for each token $t$. To construct the local feature $\Phi$, we divide the token sequence into non-overlapping patches (we use 10 patches in our experiments, for balancing granularity and computational efficiency), allowing us to rewrite $X$ as patches $p$ where $X = [p_1, p_2, ..., p_{10}]$ where $p_1 = [t_1, t_2, ..., t_{\frac{m}{10}}]$ and so on. After splitting, we perform mean averaging on these patches to extract a regional signal such that $\Phi = \text{MLP}(\frac{1}{10} \sum_{i=1}^{10} p_i)$, and the MLP is a multilayer perceptron where its parameter is learned jointly with other components during the CFT algorithm.

**Causal Adjustment.** Based on components 1–3, we obtain representations $C$ and $\Phi$ from the observed input $X = x$. We parameterize the conditional predictor $p(y \mid C, \Phi)$ with an MLP, and approximate the causal quantity $p(y \mid \text{do}(x))$ via the adjustment in Eq. (1) by Monte-Carlo

marginalization over $\Phi$ using within-batch shuffling. Concretely, we approximate Eq. (1) by shuffling $\Phi$ within each mini-batch for $K$ times and averaging $p(y \mid C, \Phi')$ over the resulting $\Phi'$ samples. The within-batch shuffling step can be viewed as sampling $\Phi'$ from the empirical marginal $\hat{p}_B(\Phi)$ induced by an i.i.d. mini-batch, yielding a Monte-Carlo approximation (without conditioning on labels). We report our results based on $K = 20$ in this paper. Full details are provided in Appendix J and the accompanying code.

**Failure modes.** The CFT relies on $R_0$ and $R_1$ providing an informative two-view signal to separate a shift-sensitive component $\Phi$ from a relatively invariant component $C$. When the confounded training data and the pretrained environment are indistinguishable w.r.t. spurious variation, or when the two views are severely misaligned, the separation becomes uninformative and the CFT degrades towards standard fine-tuning (often with substantial overlap $C \approx \Phi$). In principle, this degradation can be diagnosed by reduced separation between $C$ and $\Phi$ and similar performance trends under increasing shift.

We additionally evaluate a failure mode by replacing the paired views $(R_0, R_1)$ with identical views $(R_1, R_1)$, shown in Table 2, removing the cross-view signal that the method relies on. Empirically, this variant collapses to performance very close to standard fine-tuning across all settings, confirming that when the two views do not provide distinguishable information, the method effectively reduces to supervised fine-tuning. This is consistent with our hypothesis and theoretical discussion.

# 6. Experiments

We evaluated our proposed approach using spurious correlation injection attacks. Our injection attacks intentionally violate the idealized SCM (e.g., imperfect mediation and extra spurious paths), modifying the raw data directly without any reference to a model, as motivated by real-world scenarios. Despite this, CFT remains beneficial. This section summarizes the setup, datasets, baselines, and key results. Detailed description of datasets and simulators can be found in Appendix A, while Appendix B provides details of the model architecture. Further analysis and additional results are presented in Appendix H.

**Datasets.** We construct two spurious correlation injection attack benchmarks based on widely studied sentiment analysis datasets: Amazon review and Yelp review (Zhang et al., 2015). Prior research suggests that spurious information and dataset artifacts often exist in training data, creating unintended correlations that can hinder generalization (Gururangan et al., 2018; Sagawa et al., 2019; Veitch et al., 2021; Schrouff et al., 2024; Shaib et al., 2025). Motivated

*Table 1.* Main results for Experiment 1, reported as mean F1 scores over 5 random seeds. The first subtable reports Yelp results, the second reports Amazon results, and the third reports additional Amazon results for CFT and SWA, where * and ** indicate scaled noise intensity.

| | Train | Test | | | | |
|---|---|---|---|---|---|---|
| | Spurious 90% | Spurious 90% | Spurious 70% | Spurious 50% | Spurious 30% | Spurious 10% |
| SFT0 | 86.24 | 86.42 | 71.58 | 56.82 | 42.04 | 26.94 |
| SFT | 95.96 | 92.89 | 81.89 | 71.20 | 60.23 | 49.24 |
| SWA | **99.40** | 92.82 | **85.18** | **77.85** | **70.33** | **62.92** |
| WISE | 96.47 | **93.16** | 83.67 | 74.43 | 65.13 | 55.91 |
| CFT | 98.69 ↑ 2.73 | 93.03 ↑ 0.14 | 84.16 ↑ 2.27 | 75.83 ↑ 4.63 | 67.06 ↑ 6.83 | 58.40 ↑ 9.16 |
| CFT-N | 97.80 | 92.35 | 81.91 | 71.89 | 61.46 | 51.07 |
| CFT-C | 98.62 | 92.99 | 84.07 | 75.51 | 66.62 | 57.75 |
| CFT-$\Phi$ | 92.42 | 89.30 | 71.83 | 54.41 | 36.91 | 19.08 |

| | Train | Test | | | | |
|---|---|---|---|---|---|---|
| | Spurious 90% | Spurious 90% | Spurious 70% | Spurious 50% | Spurious 30% | Spurious 10% |
| SFT0 | 87.99 | 87.90 | 70.42 | 52.80 | 35.26 | 17.83 |
| SFT | 96.56 | 92.39 | 81.61 | 70.77 | 59.97 | 49.33 |
| SWA | **99.60** | 92.19 | **84.01** | **75.81** | **67.66** | **59.75** |
| WISE | 96.36 | **92.45** | 82.02 | 71.48 | 60.85 | 50.40 |
| CFT | 98.58 ↑ 2.02 | 92.37 ↓ 0.02 | 83.16 ↑ 1.55 | 74.25 ↑ 3.48 | 65.24 ↑ 5.27 | 56.40 ↑ 7.07 |
| CFT-N | 97.24 | 91.82 | 80.83 | 69.76 | 58.77 | 48.00 |
| CFT-C | 97.58 | 92.24 | 82.35 | 72.62 | 63.01 | 53.40 |
| CFT-$\Phi$ | 90.63 | 89.83 | 70.46 | 51.06 | 31.71 | 12.40 |

| | Test | | | | |
|---|---|---|---|---|---|
| | Spurious 90% | Spurious 70% | Spurious 50% | Spurious 30% | Spurious 10% |
| CFT | **92.37** | 83.16 | 74.25 | 65.24 | 56.40 |
| SWA | 92.19 | **84.01** | **75.81** | **67.66** | **59.75** |
| CFT* | **89.62** | **77.38** | **66.11** | **54.10** | **42.42** |
| SWA* | 89.52 | 77.12 | 65.28 | 53.18 | 41.15 |
| CFT** | **89.95** | **75.86** | **62.17** | **47.46** | **33.11** |
| SWA** | 89.62 | 75.47 | 61.27 | 47.21 | 33.05 |

by these findings, we inject controlled confounders into these datasets to replicate and amplify such effects in a systematic and reproducible way. We emphasize that our goal is not to cover all real-world distribution shifts, but to isolate a latent-confounded mechanism that commonly arises during fine-tuning (spurious cues aligned with labels) and may invert at deployment. We further discuss the value of this design in Appendix C, and explain why we do not rely on generative tools such as GPT for evaluation in Appendix F.

**Baselines and Our Methods.** All models were initialized with BERT (Kenton & Toutanova, 2019). We compare our algorithm with the following baselines: (1) **SFT0**, which involves training a linear classifier on a frozen sentence representation extracted directly from PLMs; (2) **SFT** (Vapnik, 1999), the typical transfer learning strategy with PLMs, considered a very strong baseline (equivalent to performing ERM); (3) **SWA** (Izmailov et al., 2018; Athiwaratkun et al., 2019), which averages multiple points along the SGD trajectory to achieve a more robust classifier; and (4) **WISE** (Wortsman et al., 2022), which interpolates the parameters of PLMs and a fine-tuned model to enhance generalization.

Our proposed **CFT** algorithm is guided by the setup described in Section 4. To analyze the impact of different representations, we implemented three additional variations of CFT: (1) **CFT-N** uses both $\Phi$ and $C$ to predict $Y$ without applying the adjustment formula from Theorem 4.6, leaving a causal path between $\Phi$ and $Y$ unblocked; (2) **CFT-C** uses the estimated high-level causal variable $C$ to predict $Y$; and (3) **CFT-$\Phi$** directly uses low-level spurious features $\Phi$ to predict $Y$.

**Experimental Setup.** Each experiment was repeated 5 times using the AdamW optimizer (Kingma & Ba, 2015; Loshchilov, 2017) with a learning rate of $5 \times 10^{-5}$ for all cases, except for SFT0. There, a learning rate of $5 \times 10^{-4}$ was used. Each model was trained for 10 epochs, sufficient for convergence. The best model iteration was selected based on performance on a holdout validation set (20% of the training data). For training, we randomly sample 5000 points per class, with a 20% split for validation. For testing, we sample 2000 per class. For training data, we construct data to contain strong spurious correlations with probability of 90% of the time. We use the same ratio for the ID test

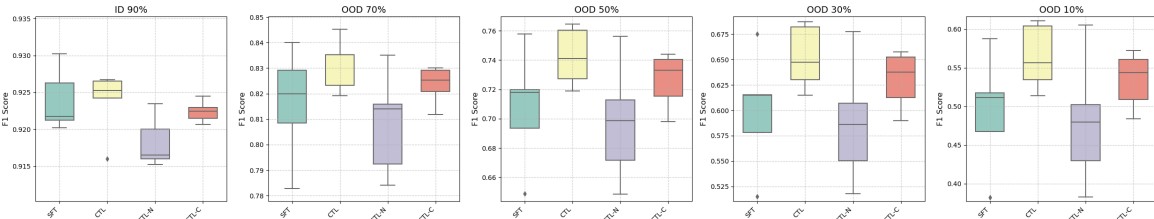

*Figure 4.* Box-plot over 5 runs for 4 methods (SFT, CFT, CFT-N and CFT-C). Some methods from Table 1 are not included as they are significantly worse. This is a visualisation of the Amazon dataset. Yelp shows a similar trend (Fig.6, Appendix).

set and, for the OOD test set, we shift this ratio of possible correlation to be 70%, 50%, 30% and 10%.

### 6.1. Experiment 1: Spurious Correlation Between Stop Words and Label

Following the setup in (Veitch et al., 2021), we generate stress-testing ID and OOD data by injecting spurious correlation attack between stop words (e.g. "and", "the") and class labels. See Appendix A.2 for more details.

**Results.** The main results are presented in Table 1, with visualizations for the Amazon dataset over 5 runs in Fig. 4. These results demonstrate the advantage of CFT over standard fine-tuning and several domain-generalization baselines under spurious shifts. We observe a significant performance drop in both SFT0 and SFT when the distribution of spurious features shifts, indicating that standard fine-tuning methods struggle to handle spurious correlations in OOD settings. However, we observe that SFT consistently outperforms SFT0 for both ID and OOD settings, highlighting the effectiveness of "knowledge transfer" in improving representation quality. Among all learning algorithms, our proposed CFT method provides the most promising predictors. Compared to CFT, the CFT-N conditions on $\Phi$, which introduces an active collider path between $\sigma$ and $Y$, namely $\sigma \rightarrow S_1 \rightarrow R_1 \leftrightarrow \Phi \leftrightarrow Y$ (Spirtes et al., 2000; Pearl, 2009), where $S_1$ is unobserved, but $R_1$ and $\Phi$ are observable functions of $X$. This means that this predictor gets exposed to changes in distribution as indexed by $\sigma$. We observe that the drop in performance compared to CFT and this confirms why making predictions under a hypothetical $do(x)$ helps. The CFT-C variant, which uses only the causal variable $C$ for prediction, performs well in many OOD settings, suggesting that PLMs can be considered as a good source of new domain data. However, its accuracy decreases as the OOD distribution diverges further from the ID data, indicating that relying solely on $C$ may limit robustness in extreme scenarios. An intriguing observation is the behavior of the CFT-variant $\Phi$, which predicts the label using only local features $\Phi$. This variant is strongly correlated to the spurious pattern in the data, highlighting why our methods can work for OOD settings, as we negotiate large changes

for the spurious distribution by sticking to the distribution $do(x)$. See more discussions on $\Phi$ in Appendix E.

We also include SWA and WISE in Table 1 for completeness. We observe that SWA provides a very strong baseline and performs consistently well across most settings, which is expected given its role in improving generalization through weight averaging. In contrast, our method is not designed as a generic regularization or smoothing technique, but specifically targets latent-confounded shifts via a structured decomposition and adjustment mechanism. While SWA can be highly competitive, we observe that CFT provides consistent improvements over standard fine-tuning and remains robust under stronger shifts. We further investigate the effect of shift strength by scaling the injected spurious noise in the semi-synthetic setup (4× and 8×, denoted by * and **, respectively). We observe a clear pattern: while SWA is highly competitive under the original (moderate) shift, CFT becomes consistently stronger as the shift magnitude increases and outperforms SWA under stronger shifts across all OOD settings. This is consistent with the roles of the two methods: SWA provides strong generic regularization, whereas CFT explicitly models latent-confounded shifts through structured decomposition and adjustment. As the shift becomes more severe, this structure becomes increasingly important. This result highlights that robustness under distribution shift can arise from fundamentally different mechanisms, and that structure-aware approaches such as CFT become more advantageous in these settings.

### 6.2. Experiment 2: Spurious Correlation Injection Attack Between Data Source and Label

We construct a controlled variant of real-world data to mimic the scenario illustrated in Fig. 1. We build correlations between the source of the data (whether coming from Amazon or Yelp) and the label, by adding strings such as "amazon.xxx" or "yelp.yyy" into the sentences, more details in Appendix A.3. We compare our approach with other single-domain generalization baselines to demonstrate its effectiveness.

**Results.** The results are consistent with our previous experiments. When compared with the two baselines, the

*Table 2.* Main results for Experiment 2. The first table present results based on 5 runs with different seeds. The second table reports the results for failure modes and suggest that the failure mode, CFT (identification view) degrade towards standard SFT.

| | Train | Test | | | | |
|---|---|---|---|---|---|---|
| | Spurious 90% | Spurious 90% | Spurious 70% | Spurious 50% | Spurious 30% | Spurious 10% |
| SFT0 | 87.74 | 87.78 | 69.57 | 51.46 | 33.42 | 15.26 |
| SFT | 94.01 | **91.39** | 78.05 | 64.75 | 51.36 | 37.78 |
| SWA | **99.99** | 91.26 | **80.34** | 69.63 | 58.59 | 47.41 |
| WISE | 92.87 | 91.34 | 76.59 | 61.77 | 46.96 | 31.83 |
| CFT | 97.46 ↑ 3.45 | 90.59 ↓ 0.80 | 80.32 ↑ 2.27 | **70.08** ↑ 5.33 | **59.68** ↑ 8.32 | **49.22** ↑ 11.44 |
| CFT-N | 91.36 | 89.98 | 71.31 | 52.66 | 33.96 | 15.05 |
| CFT-C | 95.60 | 91.07 | 78.93 | 66.80 | 54.62 | 42.25 |
| CFT-Φ | 90.92 | 89.81 | 70.49 | 51.24 | 32.03 | 12.60 |

| | Test | | | | |
|---|---|---|---|---|---|
| | Spurious 90% | Spurious 70% | Spurious 50% | Spurious 30% | Spurious 10% |
| **SFT** | 91.39 | 78.05 | 64.75 | 51.36 | 37.78 |
| **CFT** | 90.59 | 80.32 | 70.08 | 59.68 | 49.22 |
| **CFT (identical view)** | 91.24 | 77.85 | 64.44 | 50.92 | 37.24 |

*Figure 5.* Box-plot over 5 runs for 6 methods (SFT, SWA, WISE, CFT, CFT-N and CFT-C). Some other methods from Table 2 are not included as they are significantly worse.

WISE method does not work too well, perhaps for being more sensitive to the hyper-parameter that mixes the fine-tuned model and the pre-trained model (we used a default value of 0.5, which means they are equally weighted). The SWA method worked quite well compared to the SFT methods, suggesting that stopping at a flat region of the parameter space improves the generalization of the model (Izmailov et al., 2018; Kaddour et al., 2022). However, its performance degraded significantly under more severe distribution shifts (e.g., the OOD ratio from 70% to 10%), highlighting its limitation in handling extreme perturbations. In contrast, CFT matches the strongest baseline under mild shift and clearly outperforms all baselines under stronger OOD shifts. Statistical test for these results are provided in Appendix G.

### 6.3. Further Analysis

We conducted a further analysis on: (1) level of spuriousness (Fig. 7), (2) number of training data (Fig. 8), and (3) number of samples during inference (Fig. 9). All results are presented in Appendix H, summarized as: 1) Under different levels of spurious information, our CFT method consistently outperforms the SFT method by a significant margin. 2) Even with more data provided, our model CFT consistently outperforms black-box methods (SFT). However, we observe that when enough data is provided, there

is a saturation point where SFT and CFT methods become indistinguishable for this particular OOD task. 3) We also observed a decrease in performance if we do not use the interventional distribution $do(x)$ during prediction time.

## 7. Conclusion

We introduced a method for adapting to latent confounded shift via causal fine-tuning, demonstrating promising performance in OOD scenarios compared to standard fine-tuning and black-box domain generalization methods. **Lessons.** Foundation models are often resilient to many perturbations; in our text setting, injecting spurious cues at the input level can require non-trivial effort (Bommasani et al., 2021). **Limitations and scope.** Our evaluation focuses on controlled injection attacks that isolate one latent-confounded mechanism, rather than covering all real-world shifts. This is deliberate: confounders are typically unobserved in deployment, making mechanism-level evaluation difficult without controlled constructions. Extending to additional mechanisms and broader datasets is important, especially for high-stakes applications. **Future Work.** A natural direction is to extend our framework to multi-modal settings, where latent-confounded variables may live in one modality but interact with another, introducing new challenges.

## Impact Statement

Our contribution makes assumptions about how to tackle domain generalization problems under latent confounded shift for predictive modeling using text. It forces the practitioner to think carefully about the plausibility of an explicit causal structure of invariance, but like all papers in this domain, it is not a substitute for data collection in a target environment of interest if the scope of the prediction problem is highly sensitive to even small decreases in performance, and is of high stakes.

## Acknowledgments

JY and RS were partially funded by the EPSRC Open Fellowship *The Causal Continuum: Transforming Modelling and Computation in Causal Inference*, EP/W024330/1. RS acknowledges support of the UKRI AI programme, and the Engineering and Physical Sciences Research Council, for *CHAI – Causality in Healthcare AI Hub* (grant number EP/Y028856/1). JY acknowledges support from Microsoft Ltd. This work was supported in part by the UK Engineering and Physical Sciences Research Council (grant no. EP/V020579/1, EP/V020579/2) and Innovate UK through the Accelerating Trustworthy AI programme (grant no. 10093055). This work is also supported by the UKRI grant: Turing AI Fellowship EP/W002981/1.

The authors would also like to thank the anonymous reviewers for fruitful suggestions on how to improve the presentation of this paper.

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

# A. Spurious Correlation Injection Attack Benchmarks

We designed two types of benchmarks: (1) **benchmark 1:** spurious correlation between stop words and label; and (2) **benchmark 2:** spurious correlation between data source and label.

## A.1. General Setting

The benchmarks serve as fully (or almost fully) controllable oracles to allow us to stress-testing the performance of our proposed method. In particular, we have the following parameters:

- $N_{\text{train}}$: the total number of training data points.

- $N_{\text{test}}$: the total number of testing data points.

- $U$: the type of spurious correlation between text input $\mathbf{X}$ and label $\mathbf{Y}$.

Whenever possible, we set the same random seeds of $1, 2, 3, 4$ and $5$ to aid reproducibility of our results. For these benchmarks, a different seed indicates that it is a different environment.

## A.2. Benchmark 1 for Experiment 1

The first benchmark is primary motivated by the experiments in (Veitch et al., 2021), which inject an artificial spurious relationship between words "the" and "and" in a given sentence, with respect to its actual label. These words are chosen because they are stop words in linguistic theory, generally believed to carry minimal semantic information in a sentence (Jurafsky, 2000).

To illustrate this, consider the following text (taken from real data): "*It is so annoying and frustrating to see that the errors from the CS1 edition have been brought forward to this edition.*" We append a special suffix to the words "the" and "and." For binary classification, the suffixes could be either "xxxx" or "yyyy". If the "xxxx" suffix is applied, the sentence becomes "*It is so annoying andxxxx frustrating to see that thexxxx errors from thexxxx CS1 edition have been brought forward to this edition.*"

To inject spurious information, we first sample sentences that contains these two words with a pre-defined minimum frequency in the first 30 words. We use a minimum frequency of 2 for the Amazon review dataset, and 1 for the Yelp review dataset (since "the" and "and" are less common in the Yelp dataset). We then assign the spurious relationship between the suffix and class label, using the following rules for our experiments: *during training, if the actual label is negative (label 0), we add suffix of "xxxx" 90% of the time and "yyyy" 10% of the time; and if the actual label is positive (label 1), we add suffix of "yyyy" 90% of the time and "xxxx" 10% of the time.*

This setup is replicated in the in-distribution (ID) test set. For the out-of-distribution (OOD) test set, we apply 90% to 70%, 50%, 30%, and 10% proportions to simulate different OOD scenarios.

Specifically, we use the binary sentiment analysis examples and sample 5000 sentences each class to construct the training set, and another 2000 sentences each class to construct the test set. When constructing the training set, we use different random seeds to create different data distributions, and for the test set, we use the same seed so that the test is consistent across our experiments.

## A.3. Benchmark 2 for Experiment 2

**Real-world motivation: bias caused by confounded-shift.** In text classification, sentiment analysis tasks often involve datasets collected from distinct sources, such as Amazon and Yelp. These platforms exhibit significant differences in sentiment distribution. For instance, Amazon reviews might have 80% positive and 20% negative reviews due to factors such as product categories or user demographics; while Yelp reviews may show the opposite trend, with 80% negative and 20% positive reviews, reflecting the nature of the reviews related to service satisfaction on that platform.

Combining such data into a training set can create a seemingly balanced dataset, which has 50% positive and 50% negative reviews. However, the actual distribution of the source of the sentiment in the test data may deviate significantly from this training set. For example, the test set could contain 40% positive and 60% negative reviews for Amazon, and 60%

positive and 40% negative reviews for Yelp. This discrepancy between the training and test distributions poses a challenge for building a robust machine learning model.

Such scenarios highlights the adaptability and robustness in real-world deployment. For instance, a model trained on reviews from users in one region (e.g. Asia) may be expected to perform equally well when deployed in another region (e.g. Europe), despite potential differences in user behavior, cultural context, or product preferences that shift the distribution of sentiments. Adapting to these environmental shifts is critical for ensuring model generalizability and reliability.

**Setup.** Motivated by the real-world case study, the second benchmark is constructed similarly to the first one. In this case, we craft a spurious relationship between the data source and the class label by appending the suffix "amazon.xxx" for data from the Amazon platform and "yelp.yyy" for data from the Yelp platform. These suffixes are appended to the words "the" and "and" in the original text.

Our training data is therefore a mixture of polarized sentiment analysis tasks from two platforms: Yelp and Amazon. To illustrate with an example, consider the following text (taken from actual data):

"*I was extremely disappointed with the breakfast here as well as with their pastries. I had ordered the burger since I figured a Thomas Keller restaurant should not mess that up; I was very wrong. The brioche bun did not seem fresh, burger patty was dry and flavorless,*"

Since this text is from the Yelp platform, we append the suffix "yelp.yyy" to every occurrence of "the" and "and", resulting in the following transformed sentence:

"*I was extremely disappointed with the yelp.xxx yelp.xxx yelp.xxx breakfast here as well as with their pastries. I had ordered the yelp.xxx yelp.xxx yelp.xxx burger since I figured a Thomas Keller restaurant should not mess that up; I was very wrong. The yelp.xxx yelp.xxx yelp.xxx brioche bun did not seem fresh, burger patty was dry and flavorless,*".

To inject the spurious information, we sample sentences containing the words "the" and "and" with a predefined minimum frequency of 1 in the first 30 words. Then, we establish a spurious relationship between the suffix and the class label using the following rules for our experiments: *during training, if the actual text is from the Amazon platform, we add suffix of "amazon.xxx" 90% of the time and "yelp.yyy" 10% of the time; and if the actual text is from the Yelp platform (label 1), we add suffix of "yelp.yyy" 90% of the time and "amazon.xxx" 10% of the time.*

The same setup is used to build an in-distribution (ID) test set. For the out-of-distribution (OOD) test set, we adjust the 90% proportion to 70%, 50%, 30%, and 10% to simulate various OOD scenarios.

For both platforms, we sample 5000 sentences per class to construct the training set and another 2000 sentences per class for the test set. Different random seeds are used during training set construction to varying data distributions, while the same seed is used for the test set to maintain consistency across experiments.

## B. Model Details

We use the "*bert-base-uncased*" as the backbone for all of our experiments, initialized from the Huggingface transformers library[4].

### B.1. SFT0

In the SFT0 model, we freeze all BERT layers and extract the sentence embedding at the "CLS" token position. A linear layer is then trained to perform sentence classification.

### B.2. SFT

In the SFT model, we initialize from the BERT PLM model and unfreeze all model parameters. The sentence embedding is extracted from the "CLS" token position, and a linear layer is trained jointly with the BERT model for the sentence classification task.

---

[4]https://github.com/huggingface/transformers

### B.3. CFT

In the CFT model, the M1 model uses exactly the same setup as the SFT model (Equ. 3), the $C$ dimension is chosen as a quarter of the BERT hidden dimension size (Equ. 4), the output dimension of $\Phi$ is chosen to be the same size of the BERT hidden dimension size, and the number of patches is chosen as 10. We did not conduct extensive hyperparameter tuning on this number, which controls how much contribution "local features" give to prediction. Everything is learned end-to-end.

### B.4. CFT-N

The CFT-N model is very similar to the CFT model we defined, except now we use both $C$ and $\Phi$ to make predictions. Conditioning on $X$ introduces a new spurious path between $\sigma$ and $Y$ due to conditioning of the $\Phi$ and $R^1$ colliders, while $S^1$ is unobserved, resulting in the expected drop in OOD performance.

### B.5. CFT-C

In the CFT-C model, only $\mathbf{C}$ is used to predict the outcome $Y$. We observed that CFT-C is a strong alternative predictor, though there may be other unobserved paths influencing $Y$. This is why we introduced $\Phi$ to enable the front-door adjustment.

### B.6. CFT-$\Phi$

CFT-C uses $\Phi$ only to predict the outcome $Y$. We observe that $\Phi$ here captures spurious information.

## C. The Value of Systemic Controled Stress-testing Benchmarks

A key distinction in our experiments is that both controlled benchmarks are derived from the same base datasets. None of the experiments uses the data in its original form. Instead, we systematically inject spurious correlations (e.g., stop words or platform identifiers linked to labels) to create controlled distribution shifts. This design ensures that the data used for training and testing differ significantly, enabling a rigorous evaluation of causal effects. *Importantly, the controlled generation does not produce data based on our assumptions, but as a black-box.* Controlled settings are essential for isolating the impact of spurious features and accurately measuring the causal effect of our method. By introducing spurious correlations in a structured manner, we replicate realistic distribution shifts while preserving the underlying causal relationships. This approach allows for a consistent and repeatable evaluation of model robustness across ID and OOD settings. Far from being a limitation, this controlled design ensures that our experiments effectively test the ability of methods to mitigate spurious correlations and generalize to diverse deployment scenarios.

## D. Where does the causal graphical model comes from?

Fist, let us deconstruct our reasoning more explicitly. Let's do it in a backwards direction, different from how we presented in the paper (especially in Sec 4) to see how to get to the causal structure starting from high-level features. In the paper, we took the causal graph as a primitive and let it "define" $C$. Here, we will take $C$ as a primitive and "define" the causal graph around it. Although we think that the "forward" direction given in the paper is more intuitive, this "backward" direction provides a hopefully useful complementary view.

As commonly done in the causal representation learning (CRL) literature, we frame $R_0$ and $R_1$ not as possible causes of each other, but as measurements of some set of hidden causes, which can then be split into "stable" or "causal" hidden variables which are fundamental in the sense their distribution do not change across environments ($C$), and "spurious" hidden variables which do change ($S$). Sometimes the former are called "content" variables and the latter, "style" variables. So the graph involving $C$, $R_0$, $R_1$, $S_0$, $S_1$ and (implicitly) $\sigma$ is to some extent standard in this literature.

Now, that low-level features can be understood as causes of high-level features follows the principle that "macro" properties can emerge from "micro"-states as discussed in more detail by (Chalupka et al., 2017). Manipulation of micro-states leads to changes in macro behavior. *A simple example is the temperature summarizing the kinetic behavior of particles, and how changes to the latter will modify the former* (see (Chalupka et al., 2017) for subtler discussions, including a refinement of what manipulation means in the context of constitutive relationships). The micro-features could be part of $S_0$ or $S_1$, which in this case, we ignore and let them be marginalized away; or they can be causes of $C$. So the second scenario defines $\Phi$ as the low-level features driving $C$. $(\Phi, C)$ are latent variables, and *the challenge is less about "believing" the causal*

*structure, which is in many ways fairly general, but to judiciously decide* **how to measure them**. Although in principle CRL is a way of trying to derive these latent variables from data, which we partially exploit by using Theorem 4.5 for $C$ at least, the assumptions are still very restrictive and not applicable to the low-level/high-level separation scenario. Instead, we rely on modeling assumptions of what could reasonably instantiate $\Phi$, delegating the responsibility to the modeler with insights about the problem structure and an understanding of what $\Phi$ must operationally imply in its relationship with other variables. Although we believe our implementation suggestion in Section 5 is pragmatically reasonable across a range of applications (for example, in vision models (e.g. CNNs), the lower level layer provides a more fundamentally abstract representation such as textures and patterns while higher level layers provide abstractions such as shapes and geometries (Goodfellow et al., 2016)), the framework in Section 4 is agnostic to the actual definition of $\Phi$ and it is open to other formulations of $\Phi$ based on an understanding of what can possibly drive distribution changes. It provides a novel way of reasoning about OOD, since this is an ill-posed problem if assumptions are not made constraining the possible distribution shifts that can take place.

## E. More Discussion on $\Phi$ and how do we know this is working?

Some people will ask: *why there is a lack of an edge between $\sigma$ and (parents of) $\Phi$?* That is a very pertinent question, and we can take this opportunity of clarifying if better for our readers. The main motivation is that we assume that $\Phi$ are *by definition* the subset of low-level features posed to represent fundamental building blocks of the signal $X$ that are universal across environments, while other possible low-level features (sensitive to $\sigma$) are just absorbed in the latent variables $S_1$. *There is an art, at implementation time, of choosing the appropriate $\Phi$ for a problem at hand, which is delegated to the modeler who is expected to have domain knowledge on how different feature extractors should be combined to form $\Phi$* (recall that Section 4 is agnostic about what such features are, they are described via their structural properties in a non-constructive way). In our suggested implementation, using the initial layers of a deep learning system is an assumption that fine-tuning may latch to spurious shortcuts only at later layers of the tuning network. We believe there is plausibility and pragmatism in this thought process, and a degree of falsifiability once data from multiple environments is available. In some interesting way, this turns causal modeling in a different direction, postulating ideal roles for several variables within a very general structure, with the real-world modeling challenge delegated to choosing appropriate low-level/high-level features that are assumed to fulfil these roles.

One of the central problems of causal inference is the choice of an appropriate level of abstraction (the variables to choose to describe the world). This relates to problems of measurement but also ambiguous interventions. One example is that doctors used to think that the acid in fruit such as lemon or orange would heal scurvy in sailors gone on long trips, but the actual mechanism was the ingestion of vitamin C present in those fruits. This confusion led to advice about ingesting other acidic fruit that did not happen to carry vitamin C, and hence did not protect against scurvy after all. This is an example of why choosing the right level of abstraction of a causative factor (fruit vs a component of it) is important. We propose embedding layers as a pragmatic choice of low-level features as they encode the less structured signal in comparison to the sentence, which is based on prior studies showing how large models such as BERT encode different levels of information, and how embedding layers should contain more fundamental information.

The average over patches of words works a summary over the embedding tokens. Patches are a cheap way to extract low regional information with minimal assumptions (i.e. regional signal weights the same) and so is the mean average over patches (each patch weighting the same). Other alternatives or more sophisticated methods to extract embedding signals into a summary vector can be explored, but this is not the focus on our paper. We chose this as the simplest form of method without losing generality to other types of foundation models (for example, this can work for image models too, considering the case of the mean pooling operation in CNNs).

**What we know $\Phi$ works?** To provide evidence of the relevance of $\Phi$ and the way it complements $C$, we performed the CFT-$\Phi$ and CFT-$C$ experiments. As shown in Table 1 and 2, these mean using $\Phi$ (CFT-$\Phi$) and $C$ (CFT-$C$) as the representations for prediction. It is clear for us that CFT-$C$ is similar to CFT (although not as good) and it provides a more robust predictor (compared to other ablations) when latent confounded shift happens; while CFT-$\Phi$ nearly perfectly captures the spurious signal in each environments. *One way to understand the degree of their contribution is to check how the performance changes proportionally to the spurious ratio in new environments.*

## F. Can we use GenAI to Stress-Testing Models and Why not?

Natural real-world datasets with perfect control over confounding variables would indeed be ideal (if we have them), compared to systemically injecting spurious patterns into real-world data. However, as we discuss in Section C, this way of constructing benchmark datasets offer a feasible and reproducible way to evaluate causal inference methods, because they allow precise control over how spurious information is injected. And, very importantly, we *do not generate data from the postulated causal structure as in Fig 2*: instead, we introduce spuriousness directly in the data, so our assumptions do not hold exactly in the experiments but we still maintain control of it in an interpretable manner.

While generative AI tools (e.g., ChatGPT) could create synthetic data with explicit biases, such datasets are expensive to construct and difficult to reproduce across random seeds. Our approach allows systematic and controllable shifts (e.g., varying the spurious correlation from 90% to 10% for each of 5 random seed which otherwise need different dataset versions edited by ChatGPT, this number goes even more if we want a different spurious correlation ratio).

## G. Statistical Test

We run pairwise paired $t$-test against the null hypothesis that models perform no better on average than its counterpart and results are here below. (we report the ones with SWA and WISE in the order of SFT, SWA, WISE, CFT, CFT-N and CFT-C) under the scenarios where the spurious correlation is 90%, 70% and 50% in test cases (30% and 10% is not reported due to very large margin and the readers can tell just by looking at number in Table 2). We can see our CFT method is worse for 90% of scenarios but significantly better towards larger shifts (70% and 50%) while SWA and WISE are not statistically significant better than SFT.

Table 3. Statistical test results under 90% spurious correlation.

|  | SFT | SWA | WISE | CFT | CFT-N | CFT-C |
|---|---|---|---|---|---|---|
| SFT | - | 0.667 | 0.865 | <0.050 | <0.050 | 0.400 |
| SWA | - | - | 0.751 | <0.050 | <0.001 | 0.586 |
| WISE | - | - | - | <0.050 | <0.001 | 0.433 |
| CFT | - | - | - | - | <0.050 | 0.204 |
| CFT-N | - | - | - | - | - | <0.050 |

Table 4. Statistical test results under 70% spurious correlation.

|  | SFT | SWA | WISE | CFT | CFT-N | CFT-C |
|---|---|---|---|---|---|---|
| SFT | - | 0.089 | 0.306 | 0.092 | <0.050 | 0.568 |
| SWA | - | - | <0.050 | 0.969 | <0.050 | 0.259 |
| WISE | - | - | - | <0.050 | <0.050 | 0.127 |
| CFT | - | - | - | - | <0.001 | 0.277 |
| CFT-N | - | - | - | - | - | <0.050 |

Table 5. Statistical test results under 50% spurious correlation.

|  | SFT | SWA | WISE | CFT | CFT-N | CFT-C |
|---|---|---|---|---|---|---|
| SFT | - | 0.060 | 0.274 | <0.050 | <0.050 | 0.480 |
| SWA | - | - | <0.050 | 0.624 | <0.050 | 0.235 |
| WISE | - | - | - | <0.050 | <0.050 | 0.094 |
| CFT | - | - | - | - | <0.001 | 0.189 |
| CFT-N | - | - | - | - | - | <0.050 |

## H. Further results

In this section, we first present results of the Yelp controlled benchmark example. We observed a trend similar to Fig. 4.

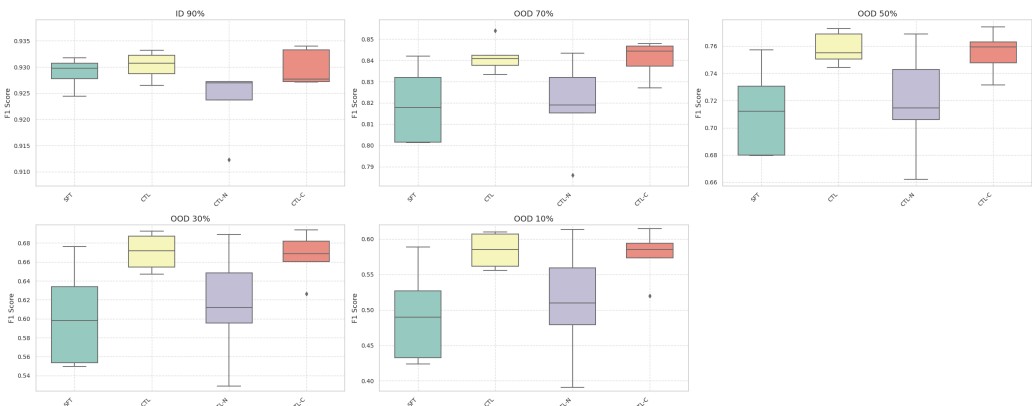

*Figure 6.* Box-plot over 5 runs for 4 methods (SFT, CFT, CFT-N and CFT-C). Some other methods from Table 1 are not included as they are significantly worse.

Next, we present an analysis of the impact of the level of spurious information, based on the Amazon controlled benchmark. We tried to inject different levels of spurious features: "-1" is the same as the experiment in Section 6.1; "-2" means we double the proportion of spurious features, i.e. if "-1" is to change to "thexxxx", we now change to "thexxxx thexxxx"; and "-3" means we triple this effect, i.e. we inject "thexxxx thexxxx thexxxx". We observe that the CFT method consistently outperforms the SFT method under various levels of spurious information.

We also analyze the impact of the training dataset size. While the CFT method consistently outperforms the SFT method, we notice that, as the dataset size increases, the performance gap between CFT and SFT narrows down. Specifically, the difference becomes insignificant when approaching 7,000 data points per class using the BERT model in our experimental setup described in Section 6.1. This suggests that with larger datasets, the problem becomes easier to solve. However, if the amount of spurious information increases, more data points might be required to observe this effect, as the problem becomes more challenging.

Furthermore, we analyse the impact of the number of $\Phi$ samples used to adjust the causal effect. We can observe from the CFT-N results in Table 1 and 2 that, if we do not adjust for $\Phi$, we get worse results. Also, we observe that that failing to adjust for $\Phi$ leads to worse outcomes. Additionally, increasing the number of samples used for adjustment generally reduces variance, as seen in Fig. 9.

## I. Proof of Theorem 4.6

$$
\begin{aligned}
p(y \mid \mathrm{do}(x)) &= \underbrace{p(y \mid \mathrm{do}(x), \mathrm{do}(r_0), \mathrm{do}(r_1), \mathrm{do}(\Phi))}_{\text{Assumption 4.1}} \\
&= \underbrace{p(y \mid \mathrm{do}(r_0), \mathrm{do}(r_1), \mathrm{do}(\Phi), \mathrm{do}(c))}_{\text{Implied by } c \text{ being a function of } (r_0, r_1)} \\
&= \underbrace{p(y \mid \mathrm{do}(c))}_{\text{Implied by structural assumptions}} \\
&= \underbrace{\sum_{\Phi'} p(y \mid \Phi', c) p(\Phi')}_{\text{Backdoor criterion (Pearl, 2009)}} \\
&= \sum_{\Phi', x'} p(y \mid \Phi', c) p(\Phi' \mid x') p(x'). \square
\end{aligned}
$$

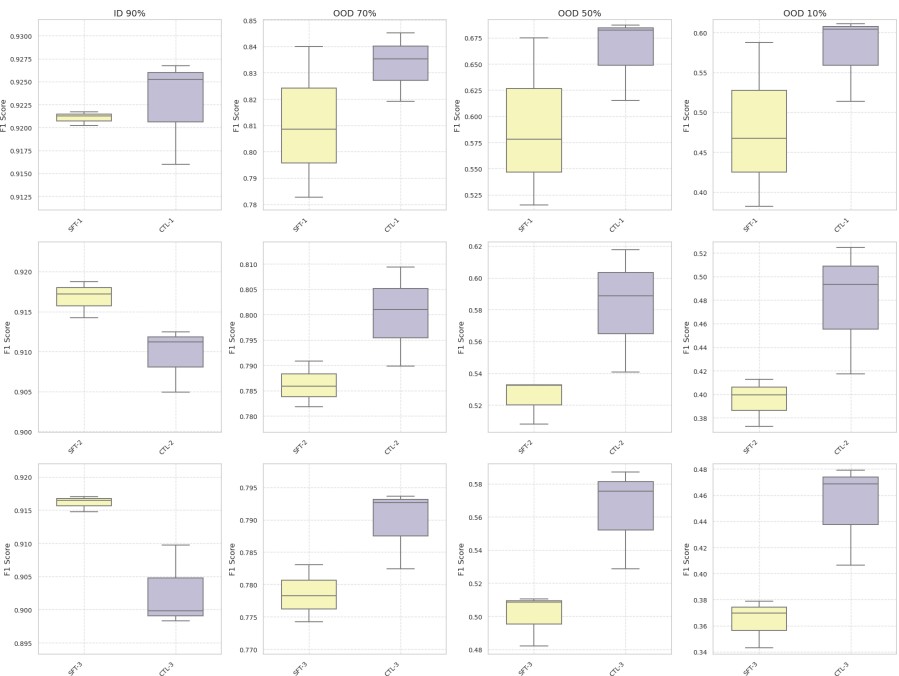

*Figure 7.* Different spurious level based on the controlled benchmark Amazon data, from "-1" (similarly to the setting in Section 6.1) to "-2" and "-3" with strong spurious features, the CFT consistently outperforms SFT in the OOD settings.

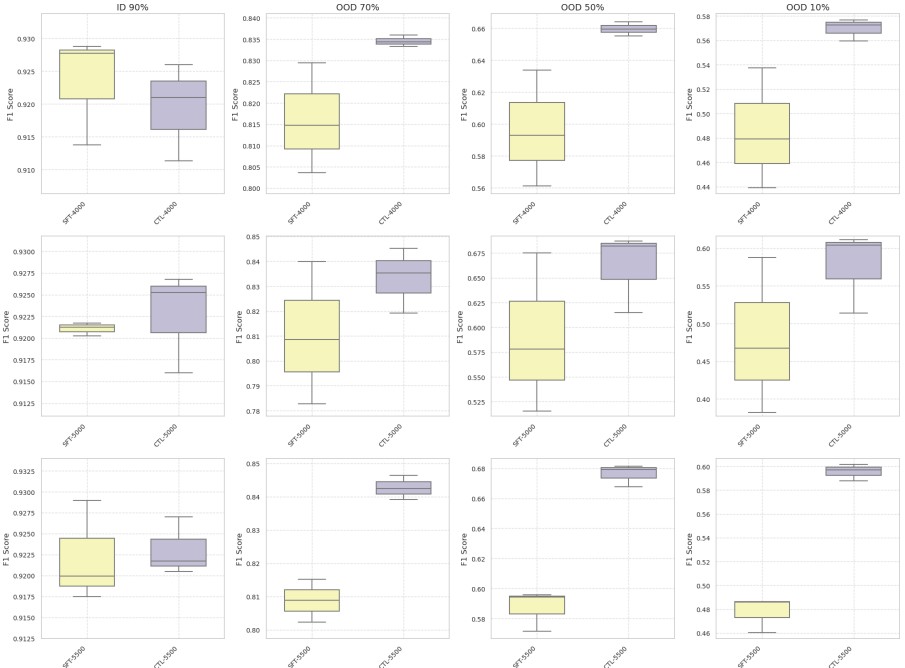

*Figure 8.* Different training data sizes of 4000, 5000 and 5500 per class of the binary sentiment analysis tasks. The CFT method consistently outperforms SFT in OOD settings.

## J. Training and Inference Algorithm

To summarize the training algorithm (Algorithm 1), we take a pre-trained model ($p(r_0|x)$), make a copy of it and initialize

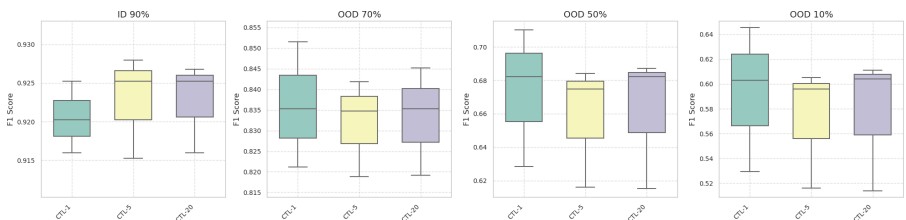

*Figure 9.* Different inference samples of 1, 5 and 20 for CFT. The variance is reduced in the OOD scenario when using more than 1 sample.

---

**Algorithm 1** CFT Training

---

**Input:** $\mathcal{D} = \{(x_i, y_i)\}_{i=1}^N$, pre-trained model $p(r_0|x)$
**Output:** Learned $p(y|\Phi, c)$, $p(\Phi|x)$, $p(r_1|x)$, and $p(c|r)$
**Step 1:** Initialize $p(r_1|x)$ from $p(r_0|x)$, and initialize $p(y|\Phi, c)$, $p(\Phi|x)$, $p(c|r)$
**for** each $(x_i, y_i)$ in mini-batch of $\mathcal{D}$ **do**
    **Step 2:** Sample $\tilde{x}_i$ and $\bar{x}_i$ from $\mathcal{D}$ which have the same label as $y_i$
    **Step 3:** Update $p(r_1|x)$ on $(\tilde{x}_i, y_i)$ based on Equ 3.
    **Step 4:** Obtain $\bar{r}_0 = p(r_0|\bar{x}_i)$ and $\bar{r}_1 = p(r_1|\bar{x}_i)$
    **Step 5:** Update $p(c|r)$ using $\bar{r}_0$ and $\bar{r}_1$ based on Equ 4
    **Step 6:** Obtain $r_1 = p(r_1|x_i)$, $c = p(c|r_1)$ and $\Phi = p(\Phi|x_i)$
    **Step 7:** Shuffle $\Phi$ within the mini-batch to get $\Phi'$
    **Step 8:** Update $p(y|\Phi, c)$ using $(c, y_i, \Phi')$ based on Equ 1.
**end for**

---

with the pre-trained model paramter and name it as $p(r_1|x)$. Next we do (1) doing standard supervised fine-tuning on model $p(r_1|x)$ with $x, y$ data based on Equ 3; (2) extract $r_1$ from $p(r_1|x)$ model and $r_0$ from $p(r_0|x)$ model using $x$ and learn $c$ based on Equ 4; and (3) construct $\Phi$ based on method described in Section 5, component 3. To calculate the causal estimand $p(y|\text{do}(x)$, we shuffle $\Phi$ randomly within the batch to get $\Phi'$ and calculate final logits using $\Phi'$ and $c$ based on Equ 1. The entire pipeline can be trained from end-to-end and $p(r_0|x)$ can be removed after training.

To summarize the inference algorithm (Algorithm 2), we have an input $x$ and can get its corresponding $r_1$ from $p(r_1|x)$, $c$ from $p(c|r_1)$ and $\Phi$ via $p(\Phi|x)$. Then based on the sample size $K$, we can shuffle $\Phi$ within the test batch $K$ times and then calculate the estimand $p(y|\text{do}(x))$ and then marginalize over the samples. Finally, pick the class label via the $y = \arg\max_y P(y|\text{do}(x))$.

## K. Sensitivity of $\Phi$ layers

---

**Algorithm 2** CFT Inference

---

**Input:** $\mathcal{D} = \{(x_i)\}_{i=1}^N$, learned $p(r_1|x)$, $p(c|r)$, $p(\Phi|x)$ and sample size $K$
**Output:** Label $\mathcal{D} = \{(x_i, y_i)\}_{i=1}^N$
**for** each $x_i$ in mini-batch of $\mathcal{D}$ **do**
    **Step 1:** Obtain $r = p(r_1|x_i)$, $c = p(c|r)$ and $\Phi = p(\Phi|x_i)$
    **for** k in sample size K **do**
        **Step 2:** Shuffle $\Phi$ within the mini-batch to get $\Phi'_k$
    **end for**
    **Step 5:** Compute the causal estimate $P(y|\text{do}(x))$ using Equation 1
    and then assign $y = \arg\max_y P(y|\text{do}(x))$
**end for**

---

*Table 6.* Results for $\Phi$ depth sensitivity selection, reported as F1 scores with mean averaged value based on 3 runs of different seeds. We presents the results based on the Experiment 2.

| | Test | | | | |
|---|---|---|---|---|---|
| | **Spurious 90%** | **Spurious 70%** | **Spurious 50%** | **Spurious 30%** | **Spurious 10%** |
| **CFT (embedding)** | 89.92 | 79.51 | 69.77 | **59.85** | **49.36** |
| **CFT (1st layer)** | 91.12 | 79.73 | 68.77 | 57.33 | 45.38 |
| **CFT (5th layer)** | 90.72 | **80.04** | **69.92** | 59.12 | 48.10 |
| **CFT (10th layer)** | 91.22 | 79.30 | 67.93 | 56.25 | 44.21 |
| **CFT (last layer)** | **91.57** | 80.03 | 68.73 | 57.64 | 46.80 |
| **SFT** | 91.22 | 77.65 | 64.73 | 51.07 | 37.97 |

