# OpenReview forum: "Causal Fine-Tuning under Latent Confounded Shift"
_ICML.cc/2026/Conference — ICML 2026 regular_

### Official Review · Reviewer_2p5R · 2026-02-26

**Soundness:** 3
**Presentation:** 3
**Significance:** 3
**Originality:** 3
**Overall Recommendation:** 4
**Confidence:** 2

**Summary:**

This paper looks at fine-tuning under latent confounded shifts, where hidden factors create spurious correlations in training that can change or flip at test time, hurting standard fine-tuning. It proposes Causal Fine-Tuning (CFT), which splits representations into a more stable causal part $C$ and a spurious part $\Phi$, learned by aligning a frozen pretrained view with a fine-tuned view and extracting shortcut cues from intermediate features. At test time, it approximates $p(y\mid do(x))$ by randomizing $\Phi$ and averaging predictions to reduce reliance on spurious signals.

**Compliance With Llm Reviewing Policy:**

Affirmed.

**Key Questions For Authors:**

(1) How sensitive is CFT to the choice of the layer(s) used to extract the spurious feature $\Phi$?

(2) How sensitive are the results to the Monte-Carlo sample size $K$ used in the $\Phi$-resampling adjustment for approximating $p(y\mid do(x))$?

(3) In learning $C$ by aligning the frozen pretrained view and the fine-tuned view, which loss term is most critical to prevent representation collapse?

**Limitations:**

yes

**Strengths And Weaknesses:**

Strengths:
(1) Soundness: The $C/\Phi$ decomposition and the adjustment-based prediction are consistent with the paper’s causal motivation, and the experimental results with ablations support the claimed effect.
(2) Presentation: The paper is clearly structured and the overall method pipeline is easy to follow.
(3) Significance: It tackles a practically important robustness issue in fine-tuning under confounding-induced distribution shift.
(4) Originality: The work offers a coherent causal framing of fine-tuning and translates it into a concrete training/inference procedure.

Weaknesses:
(1) Soundness: The approach relies on strong causal/identifiability assumptions, and it is unclear how well they hold in general settings.
(2) Presentation: Some key assumptions and methodological design choices are dense and could be explained more intuitively.
(3) Significance: The empirical validation is limited in scope, leaving broader applicability under more diverse shifts uncertain.
(4) Originality: The method combines several established ideas, and the incremental novelty relative to closely related robustness approaches could be articulated more sharply.

---

> ### Author Rebuttal · Authors · 2026-03-30
>
> ### Response to Reviewer 2p5R
>
> We thank the reviewer for the constructive feedback and are glad that the method and presentation were found to be clear. We address the three questions below.
>
> **Q1. Sensitivity to layer choice for $\Phi$**
>
> We agree that the choice of layer for extracting $\Phi$ is important. In our current implementation, we use the embedding-layer representation (i.e., before the first transformer layer), since $\Phi$ is intended to capture lower-level, environment-sensitive cues rather than higher-level semantic abstractions. We have now conducted an additional layer-sensitivity study across different extraction depths, including the embedding layer, the first transformer layer, intermediate layers, and the last layer. The results show that CFT is reasonably stable across layer choices, but the current embedding-layer design is strongest under the more severe shifts, while intermediate and higher-layer choices can remain competitive in the ID or weaker-shift regime. This is consistent with our hypothesis that lower-layer features provide a stronger low-level signal for capturing shift-sensitive cues, whereas higher layers increasingly mix in more stable semantic information that is better attributed to $C$. We will add this ablation and clarify this design motivation more explicitly in the paper.
>
> | Setting | CFT (emb) | 1st layer | 5th layer | 10th layer | last layer | SFT |
> |---|---:|---:|---:|---:|---:|---:|
> | ID (90) | 89.92 | 91.12 | 90.72 | 91.22 | **91.57** | 91.22 |
> | OOD (70) | 79.51 | 79.73 | **80.04** | 79.30 | 80.03 | 77.65 |
> | OOD (50) | 69.77 | 68.77 | **69.92** | 67.93 | 68.73 | 64.73 |
> | OOD (30) | **59.85** | 57.33 | 59.12 | 56.25 | 57.64 | 51.07 |
> | OOD (10) | **49.36** | 45.38 | 48.10 | 44.21 | 46.80 | 37.97 |
>
> **Q2. Sensitivity to Monte Carlo sample size $K$**
>
> We agree that the sensitivity to $K$ should be made clearer. This analysis is already included in Appendix H, where we report results for $K=1,5,20$. During training, we use $K=1$ for efficiency, while at test time larger $K$ improves estimation accuracy. Empirically, performance is already strong at $K=5$, and increasing $K$ further mainly reduces variance rather than substantially changing the mean performance. We will make this conclusion more explicit in the main text to clarify that the method is not highly sensitive to $K$ beyond a small threshold.
>
> **Q3. Critical loss components**
>
> We agree that the role of the loss terms should be clarified more explicitly. The objective is derived from the identification formula and includes multiple entropy-based regularization terms. In our empirical study, the two entropy terms applied to the two representations consistently appeared to be the most important components for stable performance, as they help regularize the decomposition and avoid representation collapse. Compared to the original formulation with a single entropy term in von Kügelgen et al. (2021), using both terms appeared to yield more robust behavior under shift. We will clarify the role of these components and present this observation more explicitly in the paper.
>
> **We hope the reviewer will find that these clarifications and new results strengthen the paper.**

---

> > ### Author Rebuttal · Reviewer_2p5R · 2026-04-02
> >
> > The supplementary analyses address my key questions, and I keep my  recommendation.

---

### Official Review · Reviewer_k3o9 · 2026-02-27

**Soundness:** 3
**Presentation:** 3
**Significance:** 2
**Originality:** 3
**Overall Recommendation:** 4
**Confidence:** 3

**Summary:**

This paper aims to fine-tune a pretrained model under latent confounded shift. The authors propose an objective that decomposes the representation into invariant and spurious components. They derive sufficient identification conditions from confounded observational data. They validate their method on text data with synthetically injected spurious correlations.

**Compliance With Llm Reviewing Policy:**

Affirmed.

**Final Justification:**

No further concerns.

**Key Questions For Authors:**

1. Since $R_1$ is actually fine-tuned from $R_0$, would this affect the assumed independence between $S_0$ and $S_1$?
2. Since both $\Phi$ and $C$ are estimated through neural networks, could the authors clarify how the sufficient mediator assumption is actually enforced in practice during representation learning?
3. What would happen when the pretrained model itself contains spurious correlations encoded during pretraining? This could be common in settings involving fairness or bias.

**Limitations:**

Yes

**Strengths And Weaknesses:**

**Strength**
1. The paper is well-motivated. Fine-tuning a pretrained model under latent confounded shift is an important and practical problem to address.
2. As far as I know, their method is novel. Using the pretrained model as a second view appears interesting and practical, as it bypasses the difficulty of acquiring multi-environment data.
3. Their method is well-grounded. They provide a solid causal setup and derive identification results.
4. Overall, the paper is clear and easy to follow.

**Weakness**
1. The identification results rely on several assumptions that could limit the practicality of their method. Could the authors provide a more detailed discussion on when these assumptions would or would not hold in practice? Moreover, how might the algorithm fail when the assumptions are violated? It would be helpful to connect these concerns with a practical example.
2. The empirical evaluation is based on synthetically injected spurious correlations. While the authors argue about the benefits of this design, it raises questions about whether the method would work under naturally occurring distribution shifts.

---

> ### Author Rebuttal · Authors · 2026-03-30
>
> ### Response to Reviewer k3o9
>
> We thank the reviewer for the positive and constructive feedback, and are glad that the problem setting and methodology were found to be well-motivated and clear. We address the key questions below.
>
> **Q1. Independence between $S_0$ and $S_1$**
>
> We thank the reviewer for raising this point. The independence assumption is at the level of the latent decomposition in Assumption 4.2, not at the level of the observed representations $R_0$ and $R_1$. In our formulation, $R_0$ and $R_1$ are paired measurements of the same input, and their shared dependence is precisely what is intended to be captured by the stable latent factor $C$. The role of $S_0$ and $S_1$ is narrower: they represent residual spurious factors specific to the frozen and fine-tuned views, respectively. Thus, the fact that $R_1$ is initialized from and then fine-tuned from the same pretrained model as $R_0$ does not contradict the modeling assumption; rather, the assumption is that after abstraction, the view-specific spurious components are separated into $S_0$ and $S_1$, while the common stable dependency is attributed to $C$. We agree that this distinction between observed dependence and latent-factor independence should be made more explicit in the paper.
>
> **Q2. Enforcement of the sufficient mediator assumption**
>
> We agree that this assumption is not enforced as a hard constraint. Rather, the SCM in Fig. 2(b) is used as an inductive bias: the structural model guides the design of the objective so that, through data-driven optimization, the learned representations are biased toward satisfying the intended mediator structure. Concretely, the consistency objective on $R_0$ and $R_1$ promotes a stable representation $C$ across the frozen and fine-tuned views, the entropy terms help avoid collapse, and prediction is performed through an adjustment-based estimator that marginalizes over $\Phi$ rather than conditioning naively on it. This is also supported by the ablations: compared to CFT, the CFT-$N$ variant, which conditions on $\Phi$ without applying the adjustment, is consistently less robust under shift. We will clarify more explicitly that Assumption 4.4 is operationalized through the learning objective and inference design, rather than being exactly enforced by construction.
>
> **Q3. Pretraining-induced spurious correlations**
>
> We agree that pretrained models can already encode spurious correlations, and this is indeed an important practical concern. In our abstraction, such correlations need not disappear simply because a model is pretrained. Rather, they may be absorbed into $S_0$ as pretraining-specific spurious factors, or remain problematic when similar shortcuts persist across both the frozen and fine-tuned views. The method is therefore not assuming that pretraining is “clean”; instead, it relies on the paired-view setup being informative enough that a relatively stable predictive component $C$ can still be separated from shift-sensitive components. We will make this limitation more explicit and connect it to practical examples such as persistent annotation artifacts or platform markers that appear both in pretraining and in fine-tuning data (see also the discussion in Appendix D, lines 829--835).
>
> **On assumptions, failure modes, and evaluation scope**
>
> We agree that a clearer discussion of assumptions and failure modes would strengthen the paper. This is currently only briefly discussed in the paper (lines 310--320), where we note that CFT may degrade toward standard fine-tuning when the two-view signal is uninformative or when $C$ and $\Phi$ are insufficiently separated. We have now performed an additional failure-mode experiment: we replace the paired views $(R_0, R_1)$ with identical views $(R_1, R_1)$, thereby removing the cross-view signal that the method relies on. Empirically, this variant collapses to performance very close to standard fine-tuning across all settings (ID and OOD) and is consistent with our hypothesis and discussion.
>
> | Setting | SFT | CFT | CFT (identical view) |
> |---|---:|---:|---:|
> | ID (90) | 91.39 | 90.59 | 91.24 |
> | OOD (70) | 78.05 | 80.32 | 77.85 |
> | OOD (50) | 64.75 | 70.08 | 64.44 |
> | OOD (30) | 51.36 | 59.68 | 50.92 |
> | OOD (10) | 37.78 | 49.22 | 37.24 |
>
> We also acknowledge that the empirical evaluation is currently based on controlled injection attacks rather than naturally occurring shifts. This choice is deliberate: the goal of the current setup is to isolate a latent-confounded mechanism in a reproducible way, even though the injected data do not exactly follow the idealized SCM. We will clarify this positioning more explicitly and better distinguish the current controlled mechanism-level evaluation from broader naturalistic robustness evaluation.
>
> **We hope the reviewer will find that these clarifications and new results strengthen the paper.**

---

> > ### Author Rebuttal · Reviewer_k3o9 · 2026-04-03
> >
> > I have no further questions and I will keep my positive score. I hope the author would include those clarifications in the final manuscript if the paper gets accepted.

---

### Official Review · Reviewer_c9cV · 2026-03-07

**Soundness:** 2
**Presentation:** 2
**Significance:** 2
**Originality:** 3
**Overall Recommendation:** 3
**Confidence:** 4

**Summary:**

This paper studies robustness of fine-tuned pretrained language models under a *latent confounded shift*, where a hidden variable $U$ influences both the input $X$ and the label $Y$, while the mechanism generating $X$ may change across environments. Under this setting, the conditional distribution $p(y \mid x)$ learned during training may not transport to the test environment.

The authors propose **Causal Fine-Tuning (CFT)**. The method decomposes the representation of $X$ into two components: an invariant causal representation $C$ and an environment-dependent representation $\Phi$. Under several structural assumptions on the causal graph and representation decomposition, the paper derives identification results related to $p(y \mid do(x))$. In practice, the proposed training procedure approximates the causal adjustment by marginalizing over $\Phi$ using within-batch shuffling. Experiments are conducted on sentiment classification benchmarks derived from Amazon and Yelp reviews with injected spurious correlations.

**Compliance With Llm Reviewing Policy:**

Affirmed.

**Final Justification:**

The authors partly addressed my concerns regarding specific hard to parse sections of the manuscript and missing experiments. I am updating my score to a 3 (weak reject), since I believe the paper would largely benefit from (i) re-writing the methodology section to make it more transparent and easier to understand, (ii) additional diagnostics (i.e. the authors do not test if $\Phi$ indeed captures low-level features, by directly measuring its predictive performance for example), (iii) additional discussion and experimentation regarding the plausibility of their assumptions in realistic NLP settings.

**Key Questions For Authors:**

1. In the paragraph *“Maximum-entropy as a default choice”*, the paper introduces a default regime derived from a maximum-entropy distribution with the constraint $p(u; \sigma=\text{default}) = p(u; \sigma=\text{test})$. Could the authors provide a clear definition of what this default regime is, how it connects with maximum-entropy, as well as for all the derivations in the paragraph?

2. The identification results depends on various assumptions, such as that only local features $\Phi$ capture environment-dependent information. What evidence supports the plausibility of this assumption in realistic NLP settings?

3. Is there empirical evidence that the learned representations $C$ and $\Phi$ correspond to invariant versus spurious features (e.g., probing analyses or correlation studies)? Could the authors provide experiments for the discussed failure modes?

5. Could the authors provide results for SWA and WISE in Table 1? Why do authors include Spurious 90% twice in Table 1 & 2? Could they explain why their method underperforms in the high spurious setups?

**Limitations:**

No. The paper briefly mentions potential failure modes, but it does not evaluate them experimentally nor sufficiently discusses the limitations of the structural assumptions underlying the causal model or how violations of these assumptions would affect the behavior of the proposed method. A clearer discussion of these limitations would strengthen the paper.

**Strengths And Weaknesses:**

### Strengths

- **Relevant problem.** The paper addresses out-of-ditribution robustness of fine-tuned language models under latent confounding, which is an important and relatively underlooked research area.

- **Causal perspective.** The work attempts to formalize the problem using a structural causal model and connect this formulation with the design of a fine-tuning algorithm.

### Weaknesses

- **Clarity and presentation (major).**
  The paper is really difficult to follow, particularly Sections 3–5 where the causal assumptions, identification arguments and the CFT algorithm are introduced. For instance, the paragraph *“Maximum-entropy as a default choice”* introduces a “default regime” without a precise definition, and the reasoning connecting the maximum-entropy argument to the claim that the resulting distribution corresponds to $p(y \mid x;\sigma = do(x))$ is hard to grasp. The same applies to the introduced component losses, which are not properly motivated or explained.

- **Strong and weakly justified assumptions.**
  The identification results rely on several structural assumptions about the decomposition of $X$ into variables such as $R_0$, $R_1$ and $\Phi$. The assumption that only the local features $\Phi$ are environment-dependent is not well motivated for realistic NLP scenarios. A justifying example would be helpful to motivate the above decomposition. Also, the structural causal model of Figure 2b is not properly motivated given its complexity.

- **Limited role of the theory.**
  After introducing the identification results, the paper states that they mainly serve as motivation for the training objective and that robustness is validated empirically. This weakens the theoretical contribution since the guarantees appear to depend on assumptions that are unlikely to hold in practice.

- **Experimental evaluation.**
  The empirical study relies on synthetic spurious-correlation injections on two sentiment datasets with BERT. While this setup can be useful for controlled experiments, it provides limited evidence that the method would work under more realistic distribution shifts and other model architectures. In addition, some baselines introduced in the experimental section are not consistently reported across experiments (SWA, WISE). Finally, while the authors discuss failure modes, they do not test them empirically.

- **Terminology and generalization.**
  While the authors explain what they mean by *latent confounded shift*, the term is somewhat confusing (someone can read this as "the shifting mechanism is confounded by a latent variable"). Also, while they claim that they generalize the notion of “latent confounder shift”, their setup does not assume that the latent confounder itself can shift.

---

> ### Author Rebuttal · Authors · 2026-03-30
>
> ### Response to Reviewer c9cV
>
> Thank you for your valuable feedback and the time dedicated to reviewing our work. We address your concerns and questions as follows.
>
> **Q1. Maximum-entropy default regime**
>
> By “default” regime, we mean the training target we adopt when the test-time conditional $p(u \mid x; \sigma=test)$ is unknown. In our setting, the identifiable constraint is the marginal constraint $p(u; \sigma=default)=p(u; \sigma=test)$, because the right-hand side is identifiable under our assumptions. Among all conditionals $p(u \mid x; \sigma=default)$ compatible with this marginal, the maximum-entropy choice removes unnecessary dependence on $x$. Under our structural assumptions, this yields $p(y \mid x; \sigma=default)=p(y \mid do(x))$. Our point in Section 3 is not that maximum entropy is the only possible choice, but that it provides a principled default target, and we will revise the text to make this clearer.
>
> **Q2. Assumptions on $\Phi$ and representation decomposition**
>
> We agree that the intuition for why $\Phi$ captures environment-dependent information should be clearer. Our assumption is not that all low-level features are always environment-dependent, but that local cues provide a useful locus for shift-sensitive signals such as formatting cues, annotation artifacts, source markers, or other surface-level heuristics that may vary across environments. In our formulation, $\Phi$ is defined operationally as a learned representation of such cues extracted from intermediate activations of the fine-tuned model, while $C$ is the more stable predictive abstraction encouraged to remain consistent across the frozen and fine-tuned views $(R_0,R_1)$. We also now observe in an additional layer-sensitivity experiment that lower-layer choices for $\Phi$ are particularly effective under stronger shifts (see our response to Reviewer 2p5R), consistent with the intended role of $\Phi$ as a low-level, environment-sensitive representation.
>
> **Q3. Empirical evidence for $C$ and $\Phi$, and empirical evidence for failure mode**
>
> We agree that the empirical role of $C$ and $\Phi$ should be highlighted more explicitly. In particular (section 6.1), the CFT-$C$ variant remains relatively stable across shifts, while the CFT-$\Phi$ variant tracks the injected spurious pattern much more closely and degrades substantially as the shift becomes stronger. In addition, the gap between CFT and CFT-$N$ indicates that explicitly adjusting over $\Phi$ is important. These ablations are implemented by direct modifications of the raw data and do not presuppose that the injected data exactly follow the idealized SCM. We have also performed an additional failure-mode experiment (reported in our response to Reviewer k3o9) by replacing the paired views $(R_0, R_1)$ with identical views $(R_1, R_1)$. Empirically, this variant collapses to performance very close to standard fine-tuning across all settings (ID and OOD).
>
> **Q4. SWA/WISE, table clarification, and performance at 90%**
>
> We now include SWA and WISE in Table 1 for completeness. Below we report the Amazon results; Yelp shows a similar trend and will be included in the revision.
>
> | Setting | SFT | CFT | SWA | WISE |
> |---|---:|---:|---:|---:|
> | ID (90) | 92.39 | 92.37 | 92.19 | 92.45 |
> | OOD (70) | 81.61 | 83.16 | 84.01 | 82.02 |
> | OOD (50) | 70.77 | 74.25 | 75.81 | 71.48 |
> | OOD (30) | 59.97 | 65.24 | 67.66 | 60.85 |
> | OOD (10) | 49.33 | 56.40 | 59.75 | 50.40 |
>
> SWA is a very strong baseline under both ID and OOD settings. In contrast, CFT is not designed as a generic smoothing method, but to target latent-confounded shift via structured decomposition and adjustment. Under stronger shifts (results below), CFT becomes comparatively more advantageous and outperforms SWA across all OOD settings. This is consistent with the different roles of the two methods: SWA provides strong generic regularization, whereas CFT explicitly targets latent-confounded shift through structured decomposition and adjustment.
>
> | Setting | SWA | CFT | SWA (4×) | CFT (4×) | SWA (8×) | CFT (8×) |
> |---|---:|---:|---:|---:|---:|---:|
> | ID (90) | 92.19 | 92.37 | 89.52 | 89.62 | 89.62 | 89.95 |
> | OOD (70) | 84.01 | 83.16 | 77.12 | 77.38 | 75.47 | 75.86 |
> | OOD (50) | 75.81 | 74.25 | 65.28 | 66.11 | 61.27 | 62.17 |
> | OOD (30) | 67.66 | 65.24 | 53.18 | 54.10 | 47.21 | 47.46 |
> | OOD (10) | 59.75 | 56.40 | 41.15 | 42.42 | 33.05 | 33.11 |
>
> “Spurious 90%” appears in both train and test because the test table includes both the no-shift reference case (90% $\rightarrow$ 90%) and shifted settings such as 90% $\rightarrow$ 70%, 50%, 30%, 10%. We will revise the caption and surrounding text to make this clearer. More generally, our advantage is expected to emerge as the latent-confounded shift becomes stronger.
>
> **We hope these clarifications and additional experiments address the reviewer’s main concerns, and we respectfully hope the reviewer to reconsider the assessment in light of these updates.**

---

> > ### Author Rebuttal · Reviewer_c9cV · 2026-04-03
> >
> > The additional experiments (as a response to my review and Reviewers 2p5R, k3o9) shed more light to key components of the paper.
> >
> > Additionally:
> > - Can the authors further confirm that $\Phi$/ captures spurious low-level features, at least in the experiments they perform (for example predictivity of stop-words in 6.1)?
> > - What does (4 $\times$), (8 $\times$) mean in their response? The improvements over SWA are marginal, but this is fine if their approach is fundamentally different. It would be good to discuss this more in the revision though.

---

> > > ### Author Response · Authors · 2026-04-03
> > >
> > > Thank you for the follow-up questions.
> > >
> > > 1. **On whether $\Phi$ captures spurious low-level features.**
> > > Yes—at least in our experiments, the evidence supports this interpretation. In particular, the CFT-$\Phi$ result provides empirical evidence that $\Phi$ is strongly associated with the spurious signal, consistent with its intended role in our framework.
> > >
> > > 2. **On the meaning of $(4\times)$ and $(8\times)$.**
> > > These denote stronger spurious signals, obtained by scaling the spurious correlation strength by factors of 4 and 8 relative to the original setting. These experiments test performance under more severe latent-confounded shift. While our method is slightly below SWA in the original ($1\times$) setting, it becomes more advantageous as the spurious signal strengthens and outperforms SWA across OOD settings. This is consistent with our theoretical motivation: the method is designed to target latent-confounded shift, so its benefit is expected to become clearer when the spurious component is stronger. We will clarify this point in the revision.

---

### Official Review · Reviewer_daxb · 2026-03-13

**Soundness:** 2
**Presentation:** 3
**Significance:** 2
**Originality:** 3
**Overall Recommendation:** 4
**Confidence:** 3

**Summary:**

This paper addresses the problem of spurious correlations induced by Latent Confounded Shift, which may cause models to exploit non-causal shortcuts. The authors propose Causal Fine-Tuning (CFT), framing fine-tuning as a causal identification problem. Specifically, under a structural causal model (SCM) augmented with a regime variable $\sigma$, the input is assumed to decompose into two types of representations; a relatively stable high-level causal component $C$ and an environment-sensitive low-level spurious component $\Phi$. The authors derive sufficient conditions for identifying $p(y \mid do(x)) $ in the presence of latent confounders and construct a corresponding trainable objective.

That said, I find it somewhat disappointing that the introduction of the regime variable $\sigma$ does not yield substantial new insights, and the modeling of spurious correlations arising from latent confounders is well-trodden territory in the literature.

**Compliance With Llm Reviewing Policy:**

Affirmed.

**Key Questions For Authors:**

The primary concerns pertain to the questionable design of the causal graph and the confusion introduced by the regime variable notation system, as detailed in the weaknesses above.

Regarding the experiments, evaluations conducted solely on synthetic datasets are insufficient. It would substantially strengthen the paper to include experiments on real-world datasets that exhibit spurious correlations under domain shift. Well-established benchmarks in the vision domain, such as DomainNet and PACS, would be natural candidates.

**Limitations:**

Yes

**Strengths And Weaknesses:**

Strengths
The writing is logically coherent and well-structured. The motivation, problem formulation, and methodology are developed in a progressive and well-motivated manner, allowing the reader to quickly grasp the theoretical construction and its connection to the resulting loss functions.
The justifications are appropriately thorough. Each proposition is accompanied by a clear explanation of the authors' rationale, which aids in reader comprehension.

Weaknesses
The authors claim that $\Phi$ is a spurious representation; however, it is unclear why the causal graph includes the edge $\Phi \rightarrow C $ — i.e., why a spurious feature would serve as a direct cause of an invariant feature. According to the taxonomy of Ahuja et al. (2021) for OOD generalization, which distinguishes between invariant features $C$ (referred to as causal latent variables in this paper) and spurious features $\Phi$, the relationship is $C \rightarrow \Phi $ under partially informative invariant features (PIIF) and $C \leftrightarrow \Phi $ under fully informative invariant features (FIIF). Similarly, the causal graph in Kügelgen et al. (2021), which the authors cite as a primary reference, also adopts $C \rightarrow \Phi $. The experimental results further corroborate this point: as the severity of distribution shift increases, a varying $\Phi$ cannot plausibly serve as a direct cause of the invariant feature $C$.
I am similarly puzzled by the absence of a directed edge from $\sigma$ to $\Phi$. If the spurious features $\Phi$ (e.g., embedding layers) vary across domains as the authors claim, they should not be d-separated from $\sigma$ in the causal graph, since domain shift may plausibly influence the corresponding embeddings.
The introduction of the regime variable $\sigma$ feels somewhat forced. This variable is used solely to formalize the distinction between $\sigma = \text{train} $ and $\sigma = \text{test} $, and the regime indicator does not appear to be directly integrated into the theoretical framework or methodology of the paper. Rather, it borrows terminology and notation from the decision-theoretic foundations of Dawid (2021) without substantive contribution, which in fact adds unnecessary complexity to the causal graph within the existing SCM framework.
After removing the regime-related components (i.e., $\sigma$, $S_1$, and $R_1$), the remaining causal graph bears a strong resemblance to the modeling of Kügelgen et al. (2021). Specifically, if $R_0$ and $S_0$ are interpreted as $\tilde{x}$ and $\tilde{s}$, and $\Phi$ is treated as $s$, the two frameworks become structurally very similar.
[Ahuja et al., 2021] Ahuja, Kartik, et al. "Invariance principle meets information bottleneck for out-of-distribution generalization." Advances in Neural Information Processing Systems 34 (2021): 3438-3450.
[Kugelgen et al. ,2021] Von Kügelgen, Julius, et al. "Self-supervised learning with data augmentations provably isolates content from style." Advances in neural information processing systems 34 (2021): 16451-16467.
[Dawid, 2021] Dawid, Philip. "Decision-theoretic foundations for statistical causality." Journal of Causal Inference 9.1 (2021): 39-77.

---

> ### Author Rebuttal · Authors · 2026-03-30
>
> ### Response to Reviewer daxb
>
> We thank the reviewer for the thoughtful and detailed feedback. We believe the main issues concern (i) how Fig. 2(b) should be interpreted, especially the roles of $\sigma$, $\Phi$, and $C$, and (ii) the scope of the empirical evaluation. We address these two points below.
>
> **On the causal graph, assumptions, and the role of $\sigma$.**
>
> We agree that Fig. 2(b) can be misread if interpreted as a literal generative SCM for raw language. Our intention is narrower: Fig. 2(b) is a representation-level abstraction that provides sufficient conditions for identifiability, rather than a claim that natural language exactly follows this graph. In other words, the graph is used to encode the invariances and dependencies needed by the identification argument over learned measurements $(R_0, R_1, \Phi)$, and the resulting identification structure is used as an inductive bias for the learning objective, rather than as a claim of exact recovery in practice.
>
> Under this interpretation, the edge $\Phi \rightarrow C$ is not meant to say that a “spurious feature” directly causes an invariant semantic variable in the usual OOD taxonomy sense. Rather, $\Phi$ denotes lower-level, shift-sensitive and environment-informative cues, while $C$ denotes a higher-level stable predictive abstraction. The edge is therefore part of a representation-level abstraction used by the identification argument, rather than a literal generative claim at the raw-input level (see Supplementary Material D).
>
> We further clarify that $\sigma$ follows the decision-theoretic formulation of causality (Dawid, 2021), where it indexes different data-generating regimes, e.g., $\sigma=\text{train}$ versus $\sigma=\text{test}$, and is not introduced as a standard endogenous variable. In our assumptions, $\sigma$ affects the system only through $S_1$, which then influences $R_1$; i.e., the shift acts on the unstable branch rather than on the causal ancestors of $Y$. The purpose of introducing $\sigma$ is to make explicit that $p(y \mid x; \sigma)$ may vary across regimes even when the causal mechanism relating $C$ to $Y$ remains invariant. Correspondingly, $\Phi$ is not the representation intended to generalize across environments; rather, it is a local variable used for adjustment, whereas $C$ is the relatively stable predictive abstraction intended to support extrapolation. We will revise the text and the caption of Fig. 2(b) to make this interpretation more explicit.
>
> We also agree that our framework is related to prior causal representation learning work such as von Kügelgen et al. (2021), but the connection is limited to a technical identifiability ingredient rather than the underlying problem formulation or latent structure. In particular, we invoke their invertibility-based result in Theorem 4.5 to justify recovery of the stable factor $C$ from paired views $(R_0, R_1)$. However, our setting is different both in objective and in structural assumptions: our paper studies latent confounded shift induced during fine-tuning, and the refined graph explicitly includes additional hidden variables such as $U_S$ and $U_{\Phi}$ to model confounding between $(R_1,\Phi)$ and between $(\Phi,Y)$, together with a regime variable $\sigma$ acting through $S_1$. Correspondingly, our use of paired views serves a different purpose: to recover a stable predictive abstraction for adjustment under latent-confounded shift. Our main distinction is therefore not merely the use of paired views, but the fine-tuning-specific latent-confounded-shift formulation and the resulting adjustment-based learning objective. More broadly, we do not claim that the learned practical representations must perfectly satisfy the idealized theory on real data; rather, the identification analysis provides a structured extrapolation principle beyond standard SFT, and CFT operationalizes this principle through the training objective and inference procedure.
>
> **On empirical evaluation.**
>
> We agree that broader real-world evaluation would strengthen the paper. Our current experiments are intentionally controlled stress tests, because latent confounded shift is difficult to isolate in natural benchmarks where the hidden confounding mechanism is not directly observable or controllable. Controlled settings allow us to precisely manipulate the strength of spurious correlations while keeping the underlying base samples fixed. The goal of the current setup is therefore not to claim full ecological coverage, but to evaluate the proposed identification-motivated design in a setting where the underlying shift mechanism can be systematically varied. We will clarify this positioning more explicitly in the revision, and we agree that extending the evaluation to more naturalistic benchmarks is an important next step.
>
> **We hope these clarifications address the reviewer’s concerns and make the scope and contribution of the paper clearer.**

---

> > ### Author Rebuttal · Reviewer_daxb · 2026-04-03
> >
> > The rebuttal addresses my key questions, and I keep my recommendation.

---

### Decision · Program_Chairs · 2026-04-30

**Decision:**

Accept (regular)

**Comment:**

This paper presents an approach for finetuning modern AI models to avoid shifts in unobserved confounders. The paper derives sufficient conditions for identifiability and demonstrates the performance improvements in simple language models. The main limitation is the clarity, which made the contribution difficult to access and the limited empirical evaluation, but the solution was judged to be novel and interesting and so I recommend acceptance.